# Scyphomedusae and Ctenophora of the Eastern Adriatic: Historical Overview and New Data

**Branka Pestorić [1], Davor Lučić [2,\*] , Natalia Bojanić [3], Martin Vodopivec [4] , Tjaša Kogovšek [5], Ivana Violić [2], Paolo Paliaga [6] and Alenka Malej [4]**

1   Institute of Marine Biology, University of Montenegro, 85330 Kotor, Montenegro; brankap@ucg.ac.me
2   Institute for Marine and Coastal Research, University of Dubrovnik, 20000 Dubrovnik, Croatia; ivana.violic@unidu.hr
3   Institute of Oceanography and Fisheries, 21000 Split, Croatia; bojanic@izor.hr
4   National Institute of Biology, Marine Biology Station Piran, Fornače 43, 6330 Piran, Slovenia; Martin.Vodopivec@nib.si (M.V.); Alenka.Malej@nib.si (A.M.)
5   Independent Researcher, Strunjan 125, 6320 Portorož, Slovenia; jellygist@gmail.com
6   Faculty of Natural Sciences, University of Pula, 52100 Pula, Croatia; paolo.paliaga@unipu.hr
*   Correspondence: davor.lucic@unidu.hr

**Abstract:** One of the obstacles to detecting regional trends in jellyfish populations is the lack of a defined baseline. In the Adriatic Sea, the jellyfish fauna (Scyphozoa and Ctenophora) is poorly studied compared to other taxa. Therefore, our goal was to collect and systematize all available data and provide a baseline for future studies. Here we present phenological data and relative abundances of jellyfish based on 2010–2019 scientific surveys and a "citizen science" sighting program along the eastern Adriatic. Inter-annual variability, seasonality and spatial distribution patterns of Scyphomedusae and Ctenophore species were described and compared with existing historical literature. Mass occurrences with a clear seasonal pattern and related to the geographical location were observed for meroplanktonic Scyphomedusae *Aurelia solida*, *Rhizostoma pulmo*, and to a lesser extent *Chrysaora hysoscella*, *Cotylorhiza tuberculata* and *Discomedusa lobata*. Holoplanktonic *Pelagia noctiluca* also formed large aggregations, which were seasonally less predictable and restricted to the central and southern Adriatic. Four species of Ctenophora produced blooms limited to a few areas: *Bolinopsis vitrea*, *Leucothea multicornis*, *Cestum veneris* and the non-native *Mnemiopsis leidyi*. However, differences between Adriatic subregions have become less pronounced since 2014. Our results suggest that gelatinous organisms are assuming an increasingly important role in the Adriatic ecosystem, which may alter the balance of the food web and lead to harmful and undesirable effects.

**Keywords:** jellyfish phenology; gelatinous organisms; blooms; inter-annual variability; long-term changes; Mediterranean Sea

## 1. Introduction

### 1.1. Background

Variability in jellyfish abundance, phenology, population density, and geographic distribution has been reported for many marine ecosystems worldwide [1–5]. Long-term jellyfish proliferation cycles are well known, as is considerable variation at seasonal and perennial scales [6–8]. Recent evidence suggests that jellyfish may benefit from human interactions with the oceans and increase their impact globally [3,9–12].

Jellyfish populations can form massive blooms due to life cycles favored by organisms' traits and local advective transport [13] which has also been demonstrated by experimental and modelling studies [14]. The proliferation of gelatinous macroplankton has various ecological impacts which can affect ecosystem services [15,16]. They can pose a threat to human health [17] and affect the economy. Many authors reported an increasing frequency and severity of negative impacts on marine fisheries and aquaculture over the last

50 years [18–20]. Fourteen species of Scyphomedusae and two species of Ctenophora have been associated with these adverse impacts worldwide [21]; approximately one-third of these species are also present in the Adriatic Sea.

The role of jellyfish as important consumers in the pelagic food web has long been recognized [22]. In recent decades, evidence has accumulated for various taxa, including fish and birds, that feed on gelatinous prey on a regular or episodic basis [15,23]. As jellyfish decay, organic material from the decomposing jellyfish tissue is released into the water column, promoting the establishment of a microbial food web [24,25]. A specific jellyfish-associated microbiome may influence biogeochemical cycles [26] and contain potential fish pathogens with important consequences for aquaculture [27]. Ultimately, the accumulation of jellyfish carcasses, including Scyphozoa, on the seabed can affect benthic biota through bacterial oxygen consumption and remineralization processes [28–30].

The Mediterranean Sea is considered a hotspot of marine diversity [31]. It hosts up to 18% of all known marine species, although it represents only about 0.82% of the ocean surface [32]. However, the jellyfish fauna (Scyphozoa and Ctenophora) appears to be less rich but is probably insufficiently studied compared to other taxa.

The current number of valid Scyphozoa species is the subject of ongoing debate [33], mainly due to the great morphological plasticity of these gelatinous organisms. Scyphozoan jellyfish are moderately known taxa and currently comprise about 230 species, most of which (~60%) were described between 1850 and 1950 [34]. As for Ctenophora, Mills [35] suggests that there are currently about 150–200 well-described species. Interestingly, between 1900 and 1909 twice as many ctenophore species were described than in any other decade [36]. However, Mills suggests that we now know perhaps about half of the ctenophore taxa in the sea [35].

Compared to these numbers, the Adriatic and Mediterranean seas host only a fraction of the known species. In previous studies, 13 species of Scyphozoa and 16 species of Ctenophora were recorded in the Adriatic Sea and 21 and 32, respectively, in the Mediterranean Sea. Among them, ten species of Scyphozoa and two species of Ctenophora are non-native and have been introduced into the Mediterranean and/or Adriatic Sea in recent decades. Four species and two genera of Scyphozoa have been newly described (Supplementary Materials Tables S1 and S2).

Accordingly, the objectives of this paper are: (I) to collect and systematize all available data on the occurrence of Scyphomedusae and Ctenophora species along the eastern Adriatic coast; (II) to present the most recent data on the phenology and abundance of jellyfish in the area and compare them with historical data; (III) to compare the occurrence, phenology and mass phenomena of jellyfish between subregions of the eastern Adriatic. We tested the following hypotheses: (a) Scyphomedusae and ctenophores are not uniformly represented in different subregions of the eastern Adriatic; (b) the seasonal occurrence of individual taxa is not similar in all subregions; and (c) the seasonal pattern of individual taxa has not changed over time.

*1.2. Historical Overview of Scyphozoa and Ctenophora Studies in the Adriatic Sea*

Most Scyphozoa and Ctenophora studies in the Adriatic Sea have been conducted in coastal waters and/or during periodic fishing surveys. As both gelatinous groups are extremely challenging to sample and preserve because of their fragile bodies, the application of new monitoring techniques and a broader research area, including deeper offshore waters, will likely significantly increase the number of species found in the Adriatic Sea. Among Scyphomedusae observed in the Adriatic, eight species frequently form blooms (Figure 1) compared to only four such ctenophore species (Supplementary Materials Tables S1 and S2).

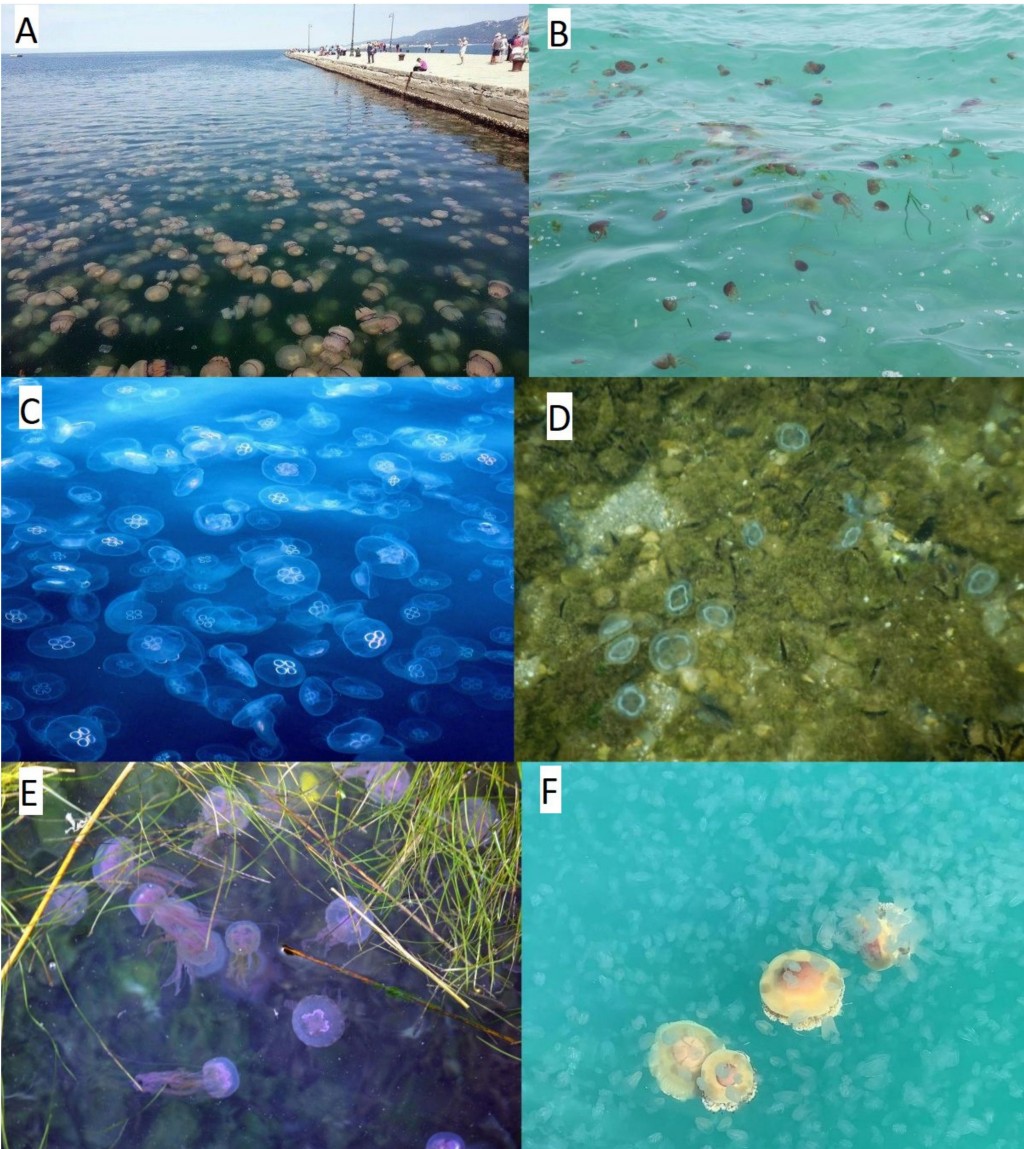

**Figure 1.** Examples of jellyfish blooms in the Adriatic Sea: (**A**) *Rhizostoma pulmo*, May 2017, Gulf of Trieste; (**B**) *Chrysaora hysoscella*, May 2017, Boka Kotorska Bay; (**C**) *Aurelia solida*, April, 2015, Gulf of Trieste; (**D**) *Discomedusa lobata*, March 2015, Boka Kotorska Bay; (**E**) *Pelagia noctiluca*, March 2007, Gulf of Trieste; (**F**) *Mnemiopsis leidyi* and *Cotylorhiza tuberculata*, September 2020, Gulf of Trieste.

Understandably, research has focused more on species that occur in large numbers and have greater impact on ecosystems and/or human activities. A glaring example is the periodic, perennial blooms of the mauve stinger *Pelagia noctiluca*, which escalated in the late 1970s–early 1980s. The blooms stimulated extensive, coordinated research efforts under the auspices of UNEP MAP (Mediterranean Action Plan) that significantly expanded knowledge of this scyphozoan species [37]. More recently, the introduction of one of the most successful invaders Ctenophora *Mnemiopsis leidyi* into Eurasian seas [14] has stimulated its study also in the Adriatic Sea [38,39].

Jellyfish research in the Mediterranean and the Adriatic Sea has a long tradition, and the basic knowledge of Scyphomedusae dates back to Aristotle as reviewed in [40]. With some of the oldest marine research stations in the world, the Adriatic Sea was an area of successful jellyfish research, especially in the second half of the nineteenth and early twentieth centuries [41]. During this period, plankton observations, including Scyphozoa and Ctenophora groups, were regularly conducted: several great biologists studied material from the Adriatic Sea and described morphology, anatomy, development and life cycles

of jellyfish [42–52]. Three species of Scyphozoa were described for the first time based on organisms found in the Adriatic Sea (the type locality); two of them in the nineteenth century: *Drymonema dalmatinum* [44] and *Discomedusa lobata* [43]. The third species, *Mawia benovici* [53], was initially described as *Pelagia benovici* [54].

With few exceptions describing sightings of some Scyphomedusae and Ctenophora species [55–57], research and reporting of jellyfish from the Adriatic stagnated between 1910 and the 1970s, and intensified again in the 1980s, especially during the last decades.

Due to massive blooms in 1977–1986, *P. noctiluca* was the most studied species in the 1980s and 1990s. Rottini Sandrini and Avian [58,59] provided new insights into the life cycle and reproduction, described the morphology of nine developmental stages, the sequence of vitellogenesis, investigated the influence of environmental factors on the reproductive period, and concluded that reproduction occurs throughout the year. Accumulations of *P. noctiluca* with 150–600 individuals per m$^3$ occurred near the north Adriatic coast, while maximum densities of medusae drifting freely in shallow water were estimated at about 20 individuals per m$^3$ [60]. Similar values in the range of 0.1–31 and >100 individuals per m$^3$ were reported by [61] for the Gulf of Trieste. Comparable maximum abundances of about 40 individuals per m$^3$ were found in the open waters of the central and southern Adriatic Sea [37]. The study of gastrovascular content indicated *P. noctiluca* as a non-selective predator feeding mainly on Copepoda and Cladocera [37,40], what was confirmed by stable isotope analysis [62]. Population dynamics studies [37] and modelling [63,64] indicated the importance of early maturation, likely favored by a food-rich environment. Besides *P. noctiluca*, a few other jellyfish species were studied at that time in the Adriatic [65] reported the bloom of *C. hysoscella* and showed cutaneous toxicity in humans.

In the following decades, jellyfish attracted more attention [66–68]. Since knowledge of species life cycle is essential for understanding bloom dynamics, researches have focused on attached polyps [69], noting that *Aurelia* polyps reached maximum abundance in summer, strobilated in the cold season, and estimated that polyps released 780–2600 × 10$^3$ ephyrae per m$^2$. Moreover, *Aurelia* polyps are capable of generating new polyps by budding, stolon production and motile bud-like tissue particles, and mode of asexual reproduction depends on their densities [70]. The three-year in situ study of the *Aurelia* polyp population [71] showed that polyp budding and stolon production were highest at temperatures above 25 °C and decreased when polyp density reached >30 polyps per cm$^2$, while strobilae were mainly formed at temperatures below 15 °C. The estimated carrying capacity K was 37.4 polyps per cm$^2$ [71]. The tedious and time-consuming work of counting tiny polyps required for population dynamic studies has been addressed by developing a new tool for automatic counting of *Aurelia* polyps [72,73]. Knowledge of some species has been greatly enhanced through the application of genetic methods and related biophysical modelling. The potential connectivity between populations of *Aurelia*, *Pelagia* and *Rhizostoma* from the Adriatic and other seas was assessed by molecular markers [74,75]. Biophysical modelling revealed the importance of artificial structures in the Adriatic as steppingstones for *Aurelia* dispersal [76]. Using an integrative approach and combining molecular and morphological analyses, Scorrano et al. [77] suggest that of the three *Aurelia* species inhabiting the Adriatic Sea, two are non-native. According to their study, the only native species *A. relicta*, which has been extensively studied in the last two decades [78–80], inhabits the marine lake on the island of Mljet.

The least known Adriatic Scyphozoa belong to the order Coronata. The small deep-sea Coronamedusae *Paraphyllina intermedia* and *Periphylla periphylla* have been recorded in the deep layers of open southern Adriatic waters, while *Nausithoe punctata* has been observed only at the surface [81,82]. *N. punctata* was regularly observed at the turn of the nineteenth to twentieth centuries in the Gulf of Trieste during the summer [45,46,83], but was later mentioned less frequently.

Recent researches examined the role of Scyphomedusae in the pelagic food webs and their impact on fisheries and aquaculture in the Adriatic Sea [19,20]. It has been

estimated that the economic losses due to a reduction in fish catches could be as high as 8.2 million euros per year for the Italian NA trawl fleet alone [20]. In their review, [21] provided convincing evidence of the significant impact of jellyfish on marine fisheries and aquaculture sectors. These include direct effects on fishing operations such as clogging and bursting of fishing nets, affecting the quantity and quality of fish caught, increased time spent sorting bycatch on board and injuries to fishers, and indirect impacts related to predation of fish eggs and larvae and changes in the food web [21].

The study of the chemical and isotopic composition of four scyphozoan species indicated the importance of sample processing before analysis [84], particularly with regard to the drying method and the effects of environmental salinities on the resulting dry mass of the jellyfish. The potential utility of jellyfish to humans, either as a source of active ingredients or as a novel food, has also been investigated [85–87]. Recent research investigated various associations of microorganisms and jellyfish [24–26].

Unlike Scyphozoa, the recent study of Ctenophora in the Adriatic has not been so productive. Until the invasion of *M. leidyi*, few publications dealt with Ctenophora. Plankton studies from the late nineteenth and early twentieth centuries [44–51,88,89] included observations of Ctenophora, while a more extensive overview of Adriatic ctenophores was provided by Krumbach [90]. Afterwards, rare observations of Ctenophora species were included in faunal lists until Mills [91] compiled the Ctenophora checklist for the Italian seas, including the Adriatic. In 2009, [92] reported on Ctenophora in the northern Adriatic and noticed the introduced species *M. leidyi* and *Beroe ovata* for the first time. *M. leidyi* has produced extensive blooms in lagoons and open waters of the northern Adriatic every summer and autumn since 2016 [38,39] and spread southwards along the Italian coast [93]. *Mnemiopsis* was rarely observed along the eastern coast of the central and southern Adriatic, whereas blooms of *Bolinopsis vitrea* were observed in the Boka Kotorska Bay, southern Adriatic, where some specimens of *B. ovata* were also found [94].

In any case, most of the jellyfish data along the eastern Adriatic coast refer to the northern Adriatic, and there is much less information for the central and southern parts. Apart from the studies on *P. noctiluca* blooms and the intensive research on *A. relicta* in the lake of Mljet Island, the only records for Scyphozoa and Ctenophora refer to the presence/absence of certain species in these regions [90] and the occurrence of some species in the period 1995–2001 [95]. We assume that the reason for this was the low interest of scientists in gelatinous macrozooplankton. Research institutions on the south-eastern Adriatic coast were established long ago: The Institute of Oceanography in Split was founded in 1930, the Oceanographic Station in Dubrovnik in 1948, and the Institute of Marine Biology in Kotor in 1961. The low interest in jellyfish stemmed from their rare occurrence and significant outbreaks. Thus, the first scientific evidence of jellyfish in the Boka Kotorska Bay was the finding of Scyphomedusae *C. hysoscella* in spring 2006 [96].

### 1.3. Study Area

The Adriatic Sea is an elongated semi-enclosed sea basin with a southeast-northwest orientation. It is the northernmost extension of the Mediterranean Sea and is relatively shallow part of the Mediterranean basin (252 m on average) with a maximum depth of 1270 m in the southern part.

The semi-enclosed nature of the entire basin (the Mediterranean Sea and its associated Adriatic Sea), combined with reduced inertia due to the relatively short residence time of its water masses, makes it highly reactive to external forces, especially to fluctuations in water, energy and matter fluxes at the interfaces which ultimately regulate the thermohaline and production properties of coastal and open waters. Freshwater inflows from the mainland, meteorological conditions, water circulation and intrusions of Mediterranean water into the Adriatic are the main factors determining physical, biogeochemical properties and biological processes in the water column.

The averaged currents at 1 m depth for the period 2010–2019 (Figure 2) show the well-known features of the Adriatic circulation [97]. Mediterranean water enters the basin

through the Otranto Straits (OS) and travels northwards along the east coast (Eastern Adriatic Current-EAC). Diluted by considerable river discharge, fresher water from the northern Adriatic travels southward along the west coast (Western Adriatic Current-WAC) towards the Ionian Sea. The EAC is weaker and less stable than WAC, but still transports jellyfish and other organisms northward toward the Gulf of Trieste (TB). The South Adriatic Gyre (SAG) and Mid Adriatic Gyre (MAG) are also visible in the image. The former is a near-permanent feature, while the latter is less stable but still persistent enough to leave a visible footprint in our 10-year average. Both contribute to cross-basin transport, affecting the transport of some jellyfish [76].

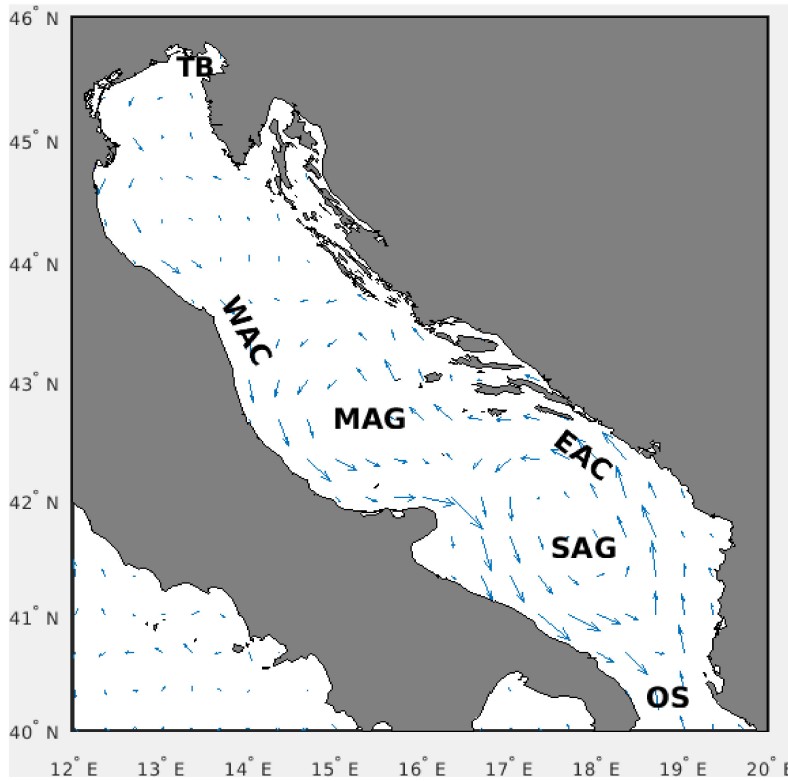

**Figure 2.** Averaged currents at 1 m depth for the 2010–2019 period. The values were taken from the Mediterranean Sea Physics Reanalysis [98] obtained through Copernicus Marine Service (CMEMS). Maximum current speed in the plot equals 0.23 m/s.

For our study we divided the coastal waters into five areas: the Gulf of Trieste (TB), the north-eastern Adriatic (NEA), the central-eastern Adriatic (CEA), the south-eastern Adriatic (SEA) and the Boka Kotorska Bay (BK) (Figure 3).

TB extends at the northernmost point of the Adriatic Sea. Because of its location and shallow depth (<30 m), it cools considerably in winter, allowing cold-affinity organisms to survive]. Sea surface temperature vary between winter minima of about 6–8 °C and late summer maxima of about 26–28 °C. The general circulation is counterclockwise, but wind-forcing greatly modulates this pattern [99]. TB is a productive area with a strong anthropogenic influence. Chlorophyll *a* biomass shows a typical seasonal pattern with mean values of 0.4–1.3 mg m$^{-3}$ and peak values in autumn [100].

NEA is closely related to the TB region, includes the western Istrian coastal waters and is roughly bounded by an isobath of 50 m depth. The Po River is the main source of freshwater and nutrients. Its annual cycle shows a bimodal pattern with peaks in late winter-spring and autumn. The NEA is a productive region with a wide range of primary production rates [101] related to the changing impact of freshwater nutrient inputs from the west coast, advection of more oligotrophic water from the central Adriatic along the east coast [100]. Chlorophyll *a* biomass shows a similar seasonal pattern as in TB.

Long-term data (1972–2009) show that in the 2000–2009 period, phytoplankton production considerably decreased, most likely caused by the low flow of the Po River [102–104]. Oligotrophication of the system is most likely caused by the low flow of the Po River and significant decrease in orthophosphate concentration, the primary limiting nutrient in the northern Adriatic.

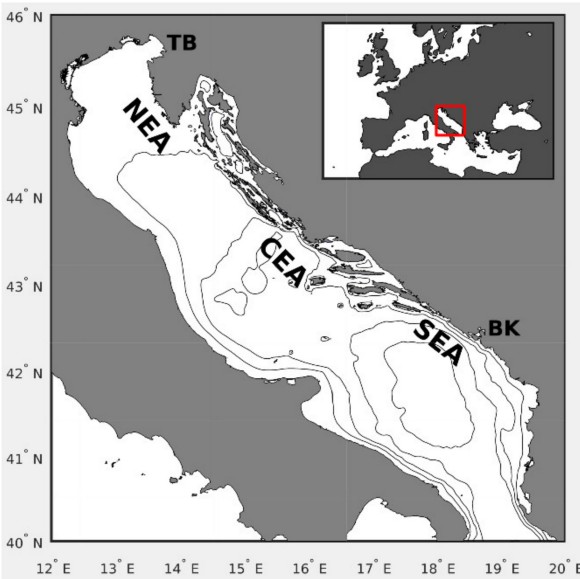

**Figure 3.** Five geographical areas where jellyfish were surveyed: TB = Gulf of Trieste, NEA = north-eastern Adriatic, CEA = central-eastern Adriatic, SEA = south-eastern Adriatic, BK = Boka Kotorska Bay. The isobaths are plotted at 50, 100, 200, and 1000 m depths.

CEA is a transition zone between the shallow northern part and the deeper southern sub-basin. The coastal area is well indented with many islands and bays. Due to their geometry and orography, each bay and channel tends to have specific oceanographic characteristics [105]. Offshore waters of CEA are oligotrophic, while productivity in the channel areas is higher than in the open sea [106]. The coast slopes rather steeply, making the channel areas relatively deep (~50 m), characterized by relatively rapid aeration, including penetration of open sea water masses. Concentration of chlorophyll *a* varies from 0.01 to 1.02 mg m$^{-3}$ [106]. On the other hand, nutrient enrichment in karst estuaries or areas affected by intensive anthropogenic pressure near large settlements and industrial zones leads to an increase in abundance and biomass of all pelagic food web components [107–109]. In the neritic area of CEA concentrations of chlorophyll *a* in the surface layer ranged from 0.01 to 7.80 mg m$^{-3}$.

SEA is a highly oligotrophic area with annual chlorophyll *a* mean value of $0.087 \pm 0.07$ mg m$^{-3}$. [110,111]. A higher level of chlorophyll *a*, ranging from 0.3 mg m$^{-3}$ to 0.7 mg m$^{-3}$, restricted to the period February–April, is short-lived and vertically limited [111,112]. The salinity in the entire water column is generally higher than 38 [113]. The changes are limited to the upper 100 m and are related to seasonal and atmospheric influences. A stronger land effect is apparent at the mouth of the Bojana River [114].

BK is a relatively large (87 km$^2$) shallow semi-enclosed area in the south-eastern part of the Adriatic Sea. Because of its extremely steep mountain walls, it is often called a fjord, although it is actually an underwater river gorge. The whole area is strongly affected by the massive inflow of freshwater from streams and underwater springs. Water exchange with the southern Adriatic mainly depends on tides, with incoming currents near the bottom and outgoing currents at the surface. Chlorophyll *a* concentrations on a 10-year scale ranged from 0.68 g m$^{-3}$ to 4.28 mg m$^{-3}$ [115]. Based on the taxonomic composition and abundance of phytoplankton, the bay is moderately eutrophic [116]. At the same time, it is under intense pressure from tourism and related urban development [117].

## 2. Materials and Methods

### 2.1. Data Collection

Data on the occurrence and abundance of Scyphozoa and Ctenophora species along the eastern Adriatic coast, in the period from 2010 to 2019, were obtained based on research cruises and surveys conducted by scientists from several Adriatic oceanographic institutions. Additional information was obtained through the Citizen Science action "Jellywatch" (leaders D. Lučić, A. Malej and B. Pestorić). Autonomous divers, in particular, provided many accurate photographs and underwater videos. Observations for each species were pooled on a monthly basis to create a multi-annual semi-quantitative data set. Each month of the year is assigned a value between 0 and 3 according to the following criteria: 0—jellyfish are not seen at all; 1—sporadic occurrence of individual organisms; 2—frequent occurrence of individual jellyfish specimens and/or small aggregations; and 3—frequent occurrence of large aggregations. The values represent the highest frequency of jellyfish occurrence in a given month, regardless of the number of reports received. Several authors have already used similar methods to present the results of macrozooplankton research [4,67,118–121].

Further, annual semi-quantitative abundance was calculated from monthly datasets using the following equation:

Σyear = [(t.o. x rel. ab. 1) + (t.o. x rel.ab. 2) + (t.o. x rel.ab. 3)]

t.o. = times observed;

rel. ab. 1, 2, 3 = relative abundance according to criteria described earlier.

### 2.2. Statistical Analyses

Semi-quantitative abundance data four Scyphozoa taxa (*Aurelia* spp., *Cotylorhiza tuberculata*, *Rhizostoma pulmo* and *Chrysaora hysoscella*) were analyzed using the non-parametric Kruskal–Wallis test and post-hoc Dunn's multiple pairwise comparison test to reveal differences in their spatial and seasonal distribution. Regarding sampling periods, samples were pooled into winter (January–March), spring (April–June), summer (July–September), and autumn (October–December) groups. The Wilcoxon signed-rank test was used to test for differences in the seasonal distribution of two taxa, Ctenophora without *Mnemiopsis leidyi* and *M. leidyi* alone.

To describe the similarity patterns of the seasonal distribution of Scyphozoa taxa, hierarchical clustering (CA) and multidimensional scaling (MDS) were used on the same data matrix, which was transformed into a lower triangular similarity matrix using the Bray–Curtis coefficients. The grouping variable "season/year" was used to extract distribution patterns in the data matrix. Clustering was performed using the group average method, applying the permutation test "similarity profile" (SIMPROOF) available in PRIMER 6 [122] to test the significance of the internal structure within the constructed clusters. An MDS plot was created to visualize the closeness of the data, where the superimposed clusters are from the cluster dendrogram and having different degrees of similarity.

Similarity percentages analysis (SIMPER) was applied to identify the jellyfish taxa that contributed most to the observed differences in seasonal or spatial distribution, using the software package PRIMER 6 [122]. The two-way analysis SIMPER for seasonal differences removed spatial differences by looking only at differences between seasons within each area and averaging this down to the contribution of taxa and vice versa. The analysis was performed using the semi-quantitative abundance matrix of selected taxa (*Aurelia* spp., *C. tuberculata*, *C. hysoscella*, *R. pulmo*, Ctenophora group and *M. leidyi*).

Principal component analysis (PCA) was applied to summarize the patterns of variation among the four Scyphozoa taxa *Aurelia* spp., *C. tuberculata*, *C. hysoscella* and *R. pulmo*, the Ctenophora group and *M. leidyi* as active variables across samples. The squared cosine values of the variables were used to estimate the best association between each variable and the extracted principal component. Varimax rotation of the extracted principal components provided better insight into the behavior of the observed variables. The analysis

was performed using the XLSTAT statistical package (version 2020.1.3, Addinsoft (2020) New York, NY, USA).

## 3. Results and Discussion

### 3.1. Taxonomic Composition, Spatial Distribution, and Relative Abundance

A complete list of species found in previous studies and this research (2010–2019) is presented in Supplementary Materials Tables S1 and S2. The results of our study yielded 10 species of Scyphozoa and nine species of Ctenophora. Since our research covered predominantly coastal areas, Coronatae species characteristic of deep waters were not observed. However, Coronatae *Nausithoe punctata* has been occasionally detected in summer in the Gulf of Trieste since 2016 [123,124]. Among other Scyphozoa, seven species previously documented in the study area and two non-native species (*Aurelia solida* and *Mawia benovici*) were confirmed. *Aurelia aurita* was recorded in the Adriatic Sea as early as the beginning of the nineteenth century [for references, see 67]. However, in view of new findings about introduced *Aurelia* species [77], we denoted this taxon as *Aurelia* spp. in our further analyses. The only exception is the newly described species *A. relicta* [77], found only in the enclosed sea lake of Mljet.

Within the Ctenophora group, we found a total of nine species, including two non-native ones: *Mnemiopsis leidyi* and *Beroe ovata* sensu Mayer [92].

Phylum CNIDARIA Verril, 1865
Subphylum MEDUSOZOA Petersen, 1979
Class SCYPHOZOA Goette, 1887

　　Order CORONATAE Van Höffen, 1902
　　Family NAUSITHOIDAE Haeckel, 1880

　　　*Nausithoe punctata* Kölliker, 1853

Subclass DISCOMEDUSAE Haeckel, 1880

　　Order RHIZOSTOMEAE Cuvier, 1800
　　Family CEPHEIDAE L. Agassiz, 1862

　　　*Cotylorhiza tuberculata* (Macri, 1778)

　　Family RHIZOSTOMATIDAE Cuvier, 1800

　　　*Rhizostoma pulmo* (Macri, 1778)

　　Order SEMAEOSTOMEAE
　　Family DRYMONEMATIDAE Haeckel, 1880

　　　*Drymonema dalmatinum* Haeckel, 1880

　　Family PELAGIIDAE Gegenbauur, 1856

　　　*Chrysaora hysoscella* (*Linnaeus, 1767*)
　　　*Mawia benovici* Avian, Ramšak, Tirelli, D'Ambra and Malej, 2016
　　　*Pelagia noctiluca* (*Forskål*)

　　Family ULMARIDAE Haeckel, 1880

　　　*Aurelia relicta* Scorrano, Aglieri, Boero, Dawson and Piraino, 2017
　　　*Aurelia solida* Browne, 1905
　　　*Discomedusa lobata* Claus, 1877

Phylum CTENOPHORA Eschcholtz 1829
Class TENTACULATA Eschscholtz 1829

　　Order CYDIPPIDA Gegenbaur, 1856
　　Family PLEUROBRACHIIDAE Chun, 1880

　　　*Pleurobrachia pileus* Müller 1776
　　　*Pleurobrachia rhodopis* Chun, 1879

　　Order LOBATA Eschscholtz 1825
　　Family BOLINOPSIDAE Bigelow 1912

> > *Bolinopsis vitrea L. Agassiz, 1860*
> > *Mnemiopsis leidyi A. Agassiz, 1865*
> Family LEUCOTHEIDAE Krumbach, 1925
> > *Leucothea multicornis* Quoy and Gaimard, 1824
> Order CESTIDA Gegenbaur, 1856
> Family CESTIDAE Gegenbaur, 1856
> > *Cestum veneris* Lesueur, 1813

Class NUDA Chun 1879

> Order BEROIDA Eschscholtz, 1825
> Family BEROIDAE Eschscholtz, 1825
> > *Beroe cucumis* Fabricius 1780
> > *Beroe forskalii* Milne Edwards, 1841
> > *Beroe ovata* Chamisso and Eysenhardt, 1821 *

* Two different animals go by this name, for a description of the taxonomic problem of *Beroe ovata* in the Adriatic see [92].

### 3.1.1. Scyphozoa

*Cotylorhiza tuberculata*

This jellyfish was observed in the warm season (Figure 4), and the frequency of observations decreased from TB to SEA. After low population densities observed from 2010–2014 along the eastern Adriatic coast, detections increased from 2015 to 2017 and then gradually decreased in 2018–2019 (Supplementary Materials Figure S3. Mass occurrences were observed very frequently in TB, especially from 2015 to 2018, while outbreaks were less frequent in NEA (Figure 4). *C. tuberculata* was continuously found in CEA, but outbreaks were only observed in August and September 2017. Bloom was not observed in SEA. In BK, *C. tuberculata* was generally rarely observed. Mass occurrence was only observed in 2013 and 2019 (Figure 4).

Since the late nineteenth century, historical records show that this species occurred in TB and NEA [42], but blooms were reported only three times until the early 1980s. Wavelet analysis indicated a six-year periodicity of higher abundance from 1978 to 2010 [67], with blooms occurring more frequently in the last decade (Figure 4). Elsewhere, there was only one record of *C. tuberculata* phenology in the SEA area until 2010 [95]. In contrast, it has recently been recorded several times in SEA in August and September (Figure 4).

The influence of environmental factors on this species in the Adriatic Sea is difficult to explain, as no polyps were found in this area. Prieto at al. [125] concluded that mass outbreaks of *C. tuberculata* in the Mar Menor are associated with a trend of global warming. Mild winters and sudden spring warming triggered strobilation, which then led to high abundance in summer, while cold winters inhibited polyp reproduction. Ruiz et al. [126] concluded that jellyfish populations fluctuate according to a simple rule: "the warmer, the better." Since *C. tuberculata* harbors autotrophic symbionts, zooxanthellae [127], food is not such an important factor in their reproduction, nor is salinity or light [125]. According to their results [125], polyp mortality after strobilation was very high (up to 100% at low temperatures), suggesting that recolonization of certain areas may occur exclusively through sexual reproduction of jellyfish. Therefore, we speculate that the occurrence of this species in SEA is dependent on recolonization from other areas.

*Rhizostoma pulmo*

This jellyfish dominated the scyphozoan community in TB, where specimens were present throughout the year (Figure 4). Overall, the frequency of occurrence and abundances peaked in 2016–2018 (Supplementary Materials Figure S3), when *R. pulmo* was also observed more frequently in other areas. In contrast to TB and NEA, the occurrence of *R. pulmo* in the southernmost regions (CEA, SEA and BK) was restricted to summer–



autumn. Large blooms (Figure 4) were observed only in TB, although *R. pulmo* was also quite common in NEA.

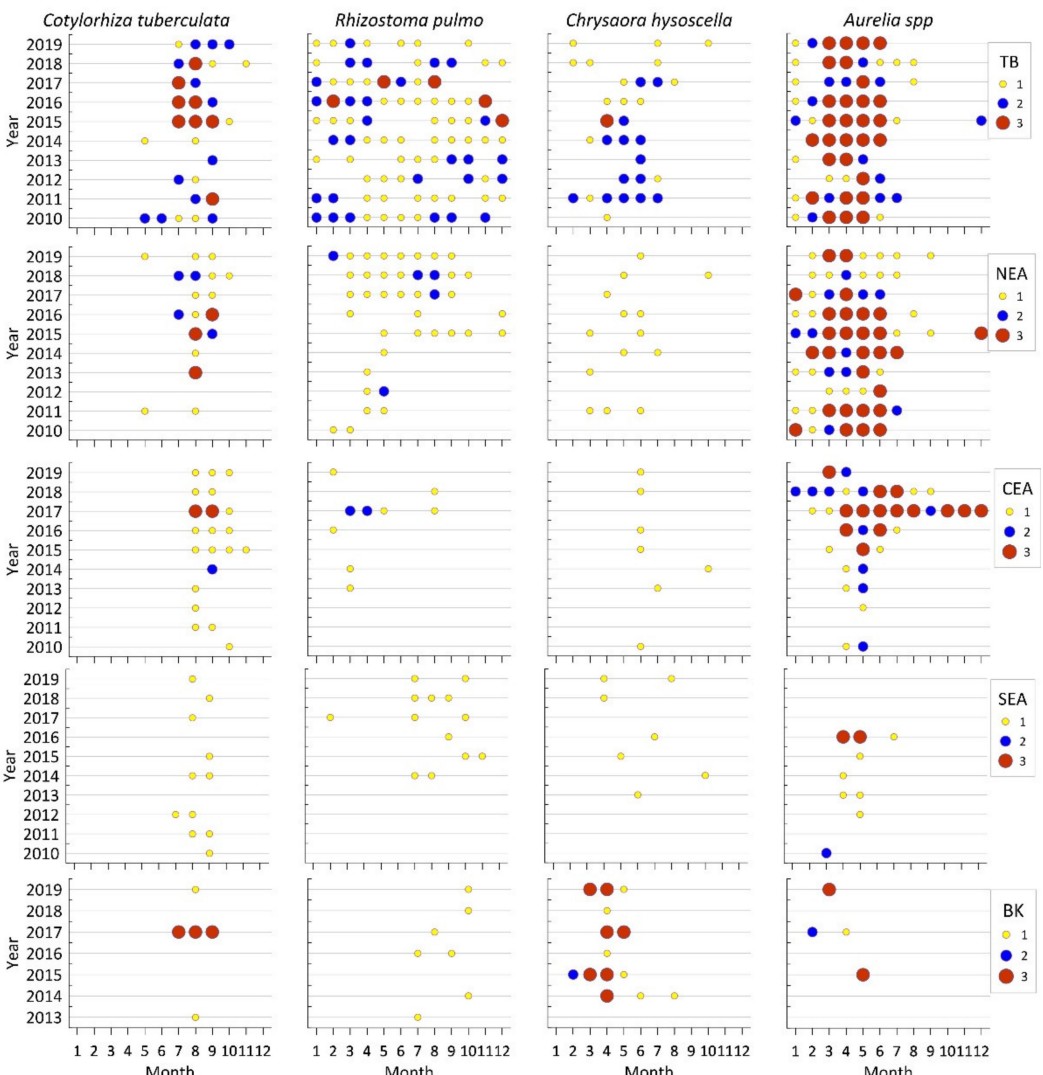

**Figure 4.** Temporal variability (month/year) of relative abundance of four dominant Scyphozoa taxa in five investigated Adriatic areas (blank—jellyfish are not seen at all; 1—sporadic occurrence of individual organisms, yellow dot; 2—frequent occurrence of individual jellyfish specimens and/or small aggregations, blue dot and 3—frequent occurrence of large jellyfish aggregations/blooms, red dot; for abbreviations see Study Area section).

Early reports [45–51,83,128,129] indicated *R. pulmo* as a common species in TB and NEA with blooms in the five years between 1899 and 1914. After that, there were no reports of outbreaks of *R. pulmo* until the first half of the 1980s. From 1980 to 2010, blooms were reported in ten years [67], while in the last decade mass accumulations were recorded in the 2015–2017 period (Figure 4). Although it was present in CEA, SEA, and BK in most years since 2013, no blooms were observed. It seems that *R. pulmo* was more abundant along the central and southern Italian coast [130]. The life cycle of *R. pulmo* includes polyps that reproduce asexually by budding and podocysts roduction, while strobilation is polydiscous [131]. To date, no polyps of this species have been found in the Adriatic Sea. A long-term study of *R. pulmo* in southern European seas [121] suggests a prolonged duration of the jellyfish season related to their earlier strobilation with increasing temperatures and possibly longer adult survival [130].

*Drymonema dalmatinum*

This species was most common in BK, where high numbers of individuals were recorded in April and May 2019, while some individuals persisted until July. It was also observed in 2014 and 2017 as well as from March to May 2020 [132]. In other areas, *D. dalmatinum* was observed only sporadically: in TB in summer 2014 and 2020, in CEA in spring 2010, 2017, 2018, and in SEA in spring 2018 (Supplementary Materials Figure S4).

*D. dalmatinum* was first described by Haeckel [44], based on a few specimens collected near island of Hvar in the CEA region. Since then, it has been rarely recorded until 1940. Based on these observations, Stiasny [133] suggested a periodicity of about 30 years. The number of *Drymonema* sightings has recently increased [134], especially in BK and TB. *D. dalmatinum* is the largest Mediterranean Scyphomedusae, a specialized predator that feeds mainly on jellyfish *Aurelia* [135,136], recently it was shown to feed also on *Rhizostoma* [137] A higher number of detections can be associated with recurrent and widespread *Aurelia* and *Rhizostoma* blooms in the Adriatic Sea in the last decades [67,68].

*Chrysaora hysoscella*

High population densities of *C. hysoscella* were found only in TB and BK in spring and early summer (Figure 4). In TB the presence of this jellyfish in the plankton was longer than in the other regions, but the bloom was only detected in 2015. In BK, the first bloom was observed in 2014 (Figure 4). In the following years, occurrences were widespread and lasted up to two months. In other study areas (NEA, CEA, SEA), *C. hysoscella* was also frequently observed, but rarely formed large aggregations (Figure 4; Supplementary Materials Figure S3).

In the northern Adriatic, *C. hysoscella* has been regularly documented since 1874 [42], but has only exceptionally reached high abundances, such as in 1989 [65]. There are only two historical records in CEA and SEA [52,88]. The first record of this jellyfish in BK dates back to 2006 [96]; subsequently it was observed in 2014. Since then, regular mass occurrences of this species have been recorded almost every year, including 2020 [132]. While no polyps were found in the wild, some of the earliest studies of *C. hysoscella* polyp morphology were conducted in the Adriatic Sea [43,52].

Although *C. hysoscella* is native to the Mediterranean, there is insufficient literature data on its distribution and abundance to compare it with other observations in the Mediterranean basins. Besides BK and TB, increased abundances of *C. hysoscella* have been recorded for the Mediterranean Sea only in the Sea of Marmara [138]. Therefore, this paper provides important information on their seasonal dynamics and records of their long-lasting outbreaks in the Mediterranean. Species of the genus *Chrysaora* are significant predators of zooplankton [139]. The number of jellyfish depends on the production of a given area [73], which is considerably higher in the northern Adriatic and BK than in other areas of the eastern Adriatic.

*Mawia benovici*

This new species, first found in the Adriatic Sea in 2013, was described as *Pelagia benovici* by [54]. A further study combining morphological and phylogenetic analyses proposed the establishment of the new genus *Mawia* [53], which was most closely related to the genus *Sanderia*. In the Mediterranean *Mawia* has so far been found only in the northern Adriatic (TB and NEA). After rare findings in autumn 2013, it was numerous in January 2014, was observed again in January 2016, October 2017 [140], December 2018, and December 2020 [123,141]. In the Adriatic it is considered a non-native species. Specimens of *Mawia benovici* were discovered off the coast of Senegal (Western Africa) by Bayha et al. [142], who suggested this as a possible region of origin for its introduction into the Adriatic Sea.

*Pelagia noctiluca*

In TB and BK, *P. noctiluca* was detected only once, in 2001 and 2018, respectively. In NEA it occurred more frequently but was represented by only a small number of specimens (Supplementary Materials Figure S5). In SEA, *P. noctiluca* occurred in all those years except 2012 and 2017, while in CEA it was recorded in 2013, 2014, 2017 and 2018 (Supplementary Materials Figures S3 and S5). Blooms occurred in spring (May–June) 2013 and April–May 2014, and in April and October 2018 in SEA. Blooms were observed in CEA in 2013 and 2018, but not in 2014. Therefore, SEA was the region with the most frequent occurrences and blooms of *P. noctiluca*.

The abundance of holoplanktonic *P. noctiluca* in different areas of the Adriatic depends on the reproduction of populations inhabiting the open southern Adriatic and on introduction from the Mediterranean Sea through the Strait of Otranto. Ephyrae of this jellyfish are often found, sometimes very abundant, in the surface layers of the open southern Adriatic basin [143]. Genetic evidence has confirmed a link between the Adriatic/Mediterranean metapopulations and the NE Atlantic [74,144].

Despite direct connection with SEA, *P. noctiluca* was observed only once in BK. During 2010–2019, *P. noctiluca* was rare in the northern areas (TB and NEA), especially compared to perennial blooms in 1978–1984 and to the mid-2000s [67]. Permanent populations have not become established in these areas [145], where the occurrence of *Pelagia* depends mainly on physical factors and high abundance of introduced individuals [64,146]. Modelling of the population dynamics of *P. noctiluca* shows that the nutrient-rich environment in the northern Adriatic stimulates the maturation of individuals of smaller size (early age), leading to a significant increase in populations [63]. However, higher mortality rates, especially in winter, prevent the establishment of viable populations. Reproduction studies of *P. noctiluca* confirm the importance of food availability and temperature [61].

*Aurelia* spp.

*Aurelia* spp. was common in all but the southernmost areas, SEA and BK (Figure 4; Supplementary Materials Figure S3). They were numerous from January to June; exceptionally in July and August. The lowest abundances and shortest period of occurrence were recorded in 2012 (Figure 4; Supplementary Materials Figure S3). *Aurelia* spp. produced blooms most frequently of all Scyphomedusae (Figure 4). In TB and NEA, mass occurrences were recorded mostly in spring, in some years for four consecutive months. An unusual bloom outside the typical season, lasting more than one month, was observed in TB in August–September 2020 [123]. In CEA, *Aurelia* spp. were present in low numbers in spring until 2015. After that, there was a significant change in phenology and abundance (Figure 4), especially around the urban area of Split. The first bloom was recorded in May 2015, while in 2017 *Aurelia* spp. were present almost all year round, with mass occurrence from spring to the end of autumn. In 2019, their occurrence and abundance decreased. (Figure 4; Supplementary Materials Figure S3). In the SEA and BK, *Aurelia* spp. was found rarely found from February to July. Episodic mass occurrences were recorded in April and May 2016 in the SEA, and in May 2015 and March 2019 in BK (Figure 4).

Since the earliest report in 1837–1838 *A. aurita* was mentioned as a common scyphozoan species in the northern Adriatic [41,67]. However, researchers recording *Aurelia* at the turn of the nineteenth and twentieth centuries [42–51,83,129] rarely reported very high abundances as opposed to blooms of *R. pulmo*. Later, *A. aurita* was observed in NEA during spring [147], while the first bloom was recorded in 1962 [41]. Until 2000, in almost 40 years, blooms were observed only in TB and NEA in 1987, 1989 and 1999. Since the mid-2000s, large aggregations of *Aurelia* occurred annually in TB and NEA [67] (Figure 4; Supplementary Materials Figure S3). After 2015 blooms became common in CEA and occasionally occurred also in SEA and BK (Figure 4).

Studies on *Aurelia* polyp populations in the Adriatic Sea indicate an important role of temperature in asexual reproduction [69,71]. The highest population density was observed in summer and is the result of lateral budding and stolon production, both of which were

observed in field studies. Under laboratory conditions, field polyps showed density-dependent reproductive modes with bud-like moving particle production at higher polyp densities [71].

### *Discomedusa lobata*

During our study, *D. lobata* was found only in TB and BK. While only a few individuals were observed in TB in summer-autumn 2017 and 2018, it was a very common jellyfish in BK with massive occurrences in winter and spring 2014 and 2015 (Supplementary Materials Figure S6). The last outbreak of this jellyfish in BK was observed in April 2020 [68].

Our results confirm the previous findings of Violić et al. [68], who referred to BK as the only area in the Mediterranean where *D. lobata* blooms regularly develop. *D. lobata* is one of the least known Scyphozoa in the Mediterranean. It was first found and described in the Gulf of Trieste in the nineteenth century [43,128], where it was described as rare in December–February [45–51,83]. A few individuals were found in NEA and CEA in 1908 and 1911 [55]. In 1981–1985, *D. lobata* was observed in winter in TB, when Avian [148] described the anatomy of this species in detail. Later, *D. lobata* was not observed in the Adriatic Sea until our findings and those presented by Violić et al. [68] for the BK.

There are sporadic records of *D. lobata* occurrence in the western Mediterranean [148–150], along the west coast of Africa [151] and in the English Channel [152]. The most recent records of *D. lobata* were from the Sea of Marmara in 2011, and with high abundances in 2013 [153]. Blooms described Violić et al. [68] in BK were the first known records of blooms for this species. Dense populations, of about 100 individuals per square meter, were found in the water column characterized by very low salinity values ranging between 8.5 and 25.5.

### 3.1.2. Ctenophora

### *Pleurobrachia pileus*

This Ctenophora was registered along the entire eastern Adriatic coast in winter-spring and autumn, always in small numbers. Most of the findings occurred in NEA.

### *Pleurobrachia rhodopis*

A few individuals were found in NEA only in March 2017 by divers.

### *Bolinopsis vitrea*

In NEA and BK, *B. vitrea* was a common native Ctenophora species in spring and autumn (Supplementary Materials Figure S7). Blooms were observed in 2014 from August to October in NEA and in October 2014 in BK. *B. vitrea* was not detected in CEA until 2013. In May 2017, the first mass occurrence of *B. vitrea* was recorded in this region. It was rarely observed in SEA, only in April 2017 and in September-October 2019. Most recently, blooms were recorded in February and March 2020 in BK [132].

### *Leucothea multicornis*

In general, abundances of *L. multicornis* increased along the eastern Adriatic coast in 2017, and it was observed in all areas in the following years (Supplementary Materials Figure S8). In NEA it was present only in February-March till 2014, afterwards it also appeared in other seasons. The bloom was recorded in June 2018 (Supplementary Materials Figure S8). In other areas, *L. multicornis* was less common. In CEA and SEA, it was recorded in summer and autumn, and in BK in winter. The only bloom outside NEA was observed in SEA in September 2012. In general, there are few data on this ctenophore for the eastern part of the Adriatic. Babić [55] documented one bloom in CEA and some individuals in SEA. Since then, until our results, there were no data for CEA, SEA and BK.

### *Bolinopsis vitrea* and *Leucothea multicornis* in TB

In TB *B. vitrea* and *L. multicornis* were not assessed separately (Figure 5) and have been

recorded continuously. High densities, but without pronounced seasonality, have occurred since 2015.

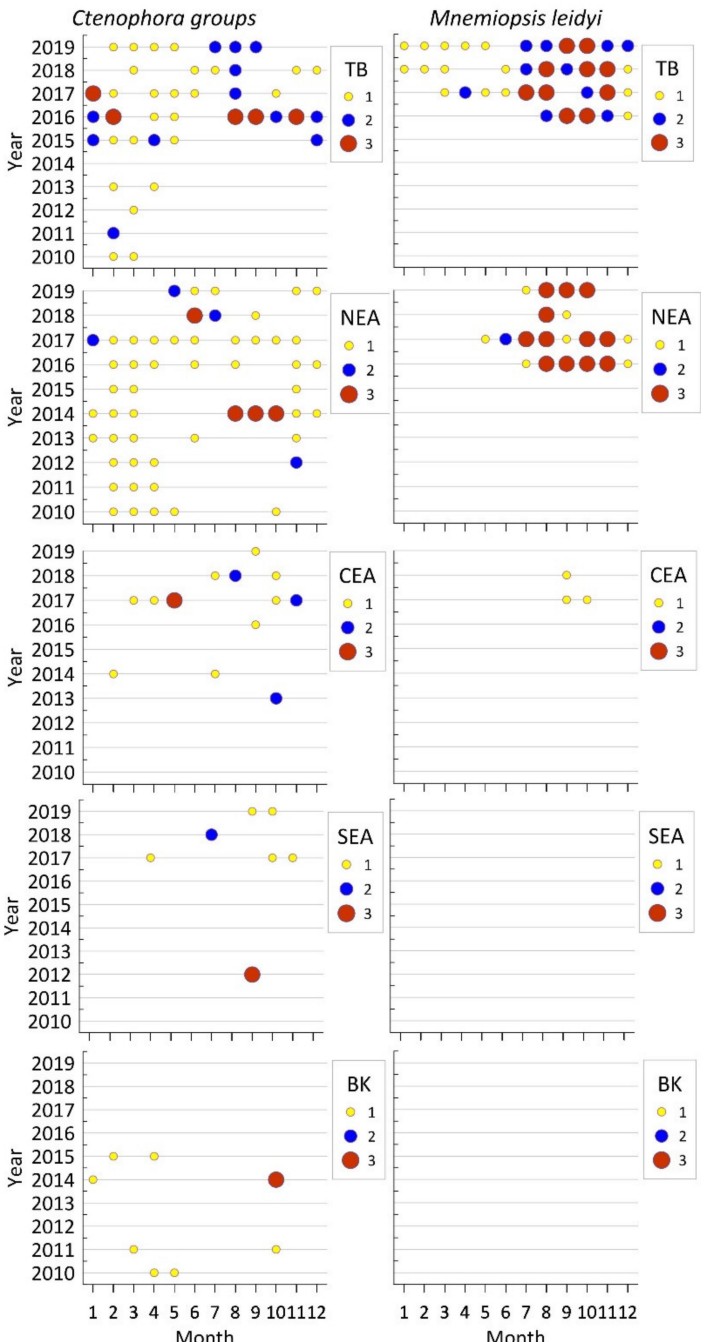

**Figure 5.** Temporal variability (month/year) of relative abundance of dominant Ctenophora taxa in five investigated Adriatic areas (blank—jellyfish are not seen at all; 1—sporadic occurrence of individual organisms, yellow dot; 2—frequent occurrence of individual jellyfish specimens and/or small aggregations, blue dot and 3—frequent occurrence of large jellyfish aggregations/blooms, red dot. Ctenophora groups stand for combined *Bolinopsis vitrea* and *Leucothea multicornis*. For abbreviations see Study Area section.

*Mnemiopsis leidyi*

*M. leidyi* was first recorded in the Gulf of Trieste in 2005 [92] but was not observed again until 2016. This invasive Ctenophora has occupied TB and NEA from summer 2016 to the present (Figure 5). It was constantly present in the plankton of TB, while in NEA

it was found in the warmer months. Large swarms of *M. leidyi* covering several square kilometers were regularly observed between July and November, reaching population densities of 300–500 individuals per m². In NEA, the occurrence of *M. leidyi* showed a general increasing trend in abundance, spatial distribution and duration of the bloom period, which continued in 2020 [154].

In other regions of the Adriatic, *M. leidyi* was found sporadically in September and October 2016 and 2017 in the port of Ploče (CEA), with a low number of individuals (Figure 5).

### *Cestum veneris*

It was a commonly present species from January to March in SEA and CEA. Blooms were recorded in February and April 2013 in SEA (Supplementary Materials S9). Swarms of this Ctenophora in the SEA were previously recorded in March and October 1999 [95]. The data show that the frequency of detections and abundances decrease from the southern region to the north. *C veneris* was sporadically found in NEA in 2015–2017, and was not observed in TB and BK.

### *Beroe forskalii*

This species was seen only in NEA, in July 2016 and September 2017 and 2019. Previously, V. Tirelli [124] mentioned the occurrence of the native *B. forskalii* and *B. ovata sensu* Chun, and introduced *B. ovata sensu* Mayer in TB. The identity of the *Beroe ovata* species is not yet clear, but it seems that the non-native *Beroe ovata*, which preys on *Mnemiopsis leidyi* in some invaded Eurasian seas [155], has not yet spread in the Adriatic.

Historically, *Leucothea (Eucharis) multicornis* was noted as a common species in the Gulf of Trieste [45–51,83], while *B. vitrea* was not mentioned at all. Instead, it was described as *Cydippe* [45–51,83]. The latter author wrote (*Pleurobrachia*) in parentheses after *Cydippe*. An earlier report on Coelenterata (Cnidaria and Ctenophora) from the Gulf of Trieste [128] mentioned *Pleurobrachia rhodopis* and *L. multicornis*. In the northern Adriatic, the authority for Ctenophora [90] listed several species: *Euchlora rubra*, *Pleurobrachia rhodopis*, *Lampetia pancerina*, *Deiopea kaloktenota*, *L. multicornis* (as the most abundant species), *Cestum veneris*, and *Beroe forskalii*. Most of these species were not mentioned in later reports on Ctenophora, most likely because there were no specialists for this group.

### *3.2. Multivariate Data Analysis*

To explore the similarity levels in Scyphozoa communities on the temporal scale (year/season) abundances in all areas were subjected to hierarchical clustering and MDS ordination (Figure 6). No grouping pattern was observed for the investigated years, while it was clearly visible for the seasons. According to a SIMPROF test, the following clusters were statistically significant: the first cluster singles out winter 2012 at a similarity level of 20.98% ($\Pi = 2.5$, $p = 0.002$), the second includes autumn samples at a similarity level of 27.74% ($\Pi = 2.19$, $p = 0.002$), and the last two clusters are separated at the similarity level of 39.69% ($\Pi = 3.14$, $p = 0.003$). The third cluster consists mostly of summer samples, while the fourth cluster contains samples from the winter–spring period. The distinction of 2012 was characterized by rather low abundances of jellyfish, which may be a consequence of extremely low winter temperatures that year in the northern Adriatic [155,156].

Table 1 illustrates the variability in the seasonal distribution of selected jellyfish taxa across all the investigated areas obtained by the SIMPER test. A contrasting pattern was evident between *Aurelia* spp., ranking first in winter and spring (January–June), and *Cotylorhiza tuberculata* and *Rhizostoma pulmo* dominating in the summer (July–September) and autumn (October-December), respectively.

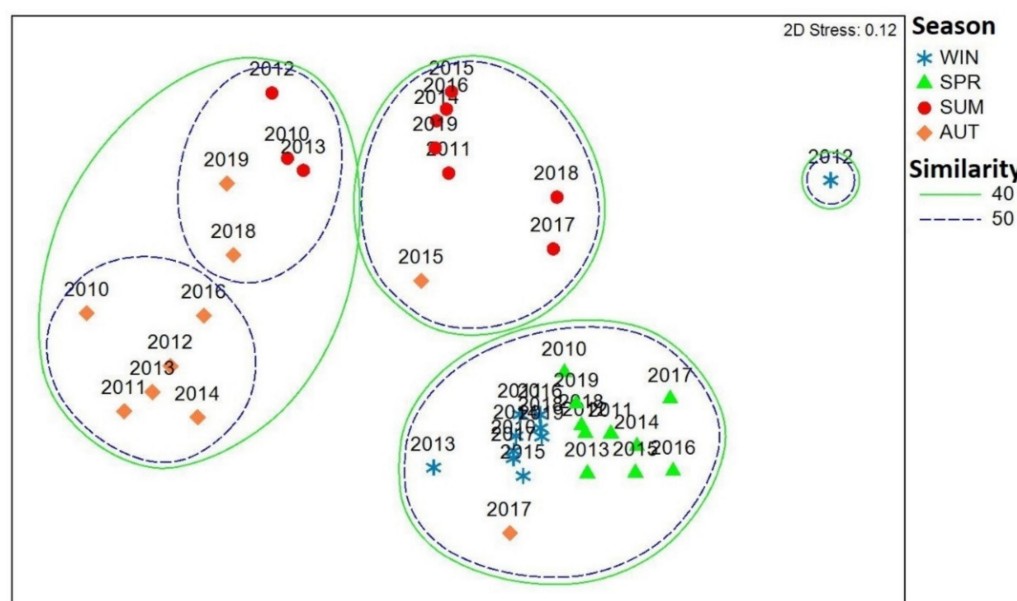

**Figure 6.** Non-metric multidimensional scaling (MDS) ordination of the sampling period (year-season) based on Bray–Curtis similarities from semi-quantitative abundance values of four Scyphozoa taxa: *Aurelia* spp., *Cotylorhiza tuberculata*, *Chrysaora hysoscella*, and *Rhizostoma pulmo.*

**Table 1.** Average contribution of four Scyphozoa taxa within each season across all area groups. (AU, *Aurelia* spp.; CT, *Cotylorhiza tuberculata*; CH, *Chrysaora hysoscella*; and RP, *Rhizostoma pulmo*).

| Winter | | Spring | | Summer | | Autumn | |
|---|---|---|---|---|---|---|---|
| (av. similarity 41.36%) | | (av. similarity 53.16%) | | (av. similarity 41.52%) | | (av. similarity 22.17%) | |
| Taxa | Contrib. [%] | Taxa | Contrib. [%] | Taxa | Contrib. [%] | Taxa | Contrib. [%] |
| AU | 69.6 | AU | 69.61 | CT | 70.46 | RP | 81.37 |
| RP | 27.95 | CH | 19.65 | RP | 22.41 | CT | 18.63 |
| CH | 2.45 | RP | 10.6 | AU | 5.42 | AU | 0.00 |
| CT | 0.00 | CT | 0.14 | CH | 1.71 | CH | 0.00 |

On the other hand, *Chrysaora hysoscella* was poorly represented in this community with the highest rank in spring. A Kruskal–Wallis test performed for each taxon showed significant seasonal differences ($p < 0.0001$) between the indicated seasons as follows for *Aurelia* spp. (winter and spring; K = 142.36), *C. tuberculata* (summer; K = 176.66), and *C. hysoscella* (spring; K = 65.56), while the differences in the seasonal distribution of *R. pulmo* were not significant.

The Ctenophora group made the highest numerical contribution to the observed community composition in the autumn-winter period (K = 9.03, $p = 0.029$). Overall *M. leidyi* was ranked low regardless of season, reflecting its absence until 2016. However, significant differences were recorded between winter-spring and summer-autumn periods (K = 37.32, $p < 0.0001$), when only the period from 2016 to 2019 was considered. The differences in the seasonal distribution between *M. leidyi* and Ctenophora group for the same period are significant (Wilcoxon signed-rank test; V = 422, $p = 0.003$).

Regarding spatial distribution, SIMPER test results indicate the greatest similarity between the areas TB and NEA (average dissimilarity of 52.52%). On the other hand, greater differences in jellyfish community composition (average dissimilarity ~80%) were recorded between TB, NEA, and CEA compared to the SEA and BK (Table 2). This test also indicates that *R. pulmo* made the largest numerical contribution (from 43.55% to 52.81%) to the observed differences in community composition between TB and the other areas. On the other hand, *Aurelia* spp. contributed from 44.53% to 51.34% and from 31.94% to 40.99%, respectively, to the differences observed between NEA and CEA compared to other study

areas. *C. hysoscella* made the greatest contribution of 35.11% to the differences between SEA and BK. The detailed contributions of Scyphozoa and Ctenophora taxa to the differences in spatial distribution are shown in Table 2.

**Table 2.** Results of SIMPER analysis on spatial differences in Scyphozoa and Ctenophora distribution with respect to the selected taxa.

| Variable | Av. Diss. between Groups | Diss./SD | Contrib. [%] | Cumulative % |
|---|---|---|---|---|
| *TB vs. NEA (av. dissimilarity 55.02%)* | | | | |
| *R. pulmo* | 16.95 | 0.88 | 30.80 | 30.80 |
| *M. leidyi* | 11.23 | 0.64 | 20.40 | 51.21 |
| Ctenophora group | 9.53 | 0.86 | 17.32 | 68.52 |
| *TB vs. CEA (av. dissimilarity 76.15%)* | | | | |
| *R. pulmo* | 25.30 | 1.14 | 33.22 | 33.22 |
| *Aurelia* spp. | 16.82 | 0.95 | 22.09 | 55.31 |
| Ctenophora group | 10.55 | 0.94 | 13.86 | 69.16 |
| *TB vs. SEA (av. dissimilarity 84.57%)* | | | | |
| *R. pulmo* | 27.79 | 1.10 | 32.86 | 32.86 |
| *Aurelia* spp. | 21.24 | 1.00 | 25.12 | 57.98 |
| Ctenophora group | 10.61 | 0.88 | 12.54 | 70.52 |
| *TB vs. BK (av. dissimilarity 81.21%)* | | | | |
| *R. pulmo* | 26.16 | 1.11 | 32.22 | 32.22 |
| *Aurelia* spp. | 18.77 | 0.97 | 23.12 | 55.33 |
| *M. leidyi* | 10.14 | 0.55 | 12.48 | 67.81 |
| *NEA vs. CEA (av. dissimilarity 72.47%)* | | | | |
| *Aurelia* spp. | 22.90 | 1.01 | 31.60 | 31.60 |
| Ctenophora group | 17.61 | 0.85 | 24.30 | 55.89 |
| *C. tuberculata* | 9.99 | 0.50 | 13.79 | 69.68 |
| *NEA vs. SEA (av. dissimilarity 86.65%)* | | | | |
| *Aurelia* spp. | 29.69 | 1.10 | 34.27 | 34.27 |
| Ctenophora group | 21.24 | 0.82 | 24.51 | 58.78 |
| *R. pulmo* | 13.43 | 0.88 | 15.50 | 74.28 |
| *NEA vs. BK (av. dissimilarity 84.15%)* | | | | |
| *Aurelia* spp. | 25.88 | 1.04 | 30.76 | 30.76 |
| Ctenophora group | 19.10 | 0.76 | 22.70 | 53.45 |
| *R. pulmo* | 12.72 | 0.78 | 15.11 | 68.57 |
| *CEA vs. SEA (av. dissimilarity 80.18%)* | | | | |
| *Aurelia* spp. | 28.29 | 0.83 | 35.28 | 35.28 |
| *R. pulmo* | 16.20 | 0.59 | 20.20 | 55.48 |
| *C. tuberculata* | 14.10 | 0.52 | 17.58 | 73.07 |
| *CEA vs. BK (av. dissimilarity 86.37%)* | | | | |
| *Aurelia* spp. | 23.11 | 0.79 | 26.76 | 26.76 |
| Ctenophora group | 17.56 | 0.70 | 20.33 | 47.09 |
| *C. tuberculata* | 16.88 | 0.61 | 19.55 | 66.64 |
| *SEA vs. BK (av. dissimilarity 83.44%)* | | | | |
| *C. hysoscella* | 23.65 | 0.74 | 28.34 | 28.34 |
| Ctenophora group | 19.18 | 0.67 | 22.99 | 51.33 |
| *R. pulmo* | 17.76 | 0.58 | 21.28 | 72.61 |

Kruskal–Wallis statistics show that the spatial distribution of *Aurelia* spp. in the Adriatic is not homogeneous, and significant differences were recorded between the northern, central and southern Adriatic (K = 104.56, <0.0001). For *C. tuberculata*, statistically significant differences were recorded for SEA, including BK, with respect to TB (K = 18.85, $p < 0.001$). Statistically significant differences in the spatial distribution of *C. hysoscella* were recorded for TB compared to all other areas except BK (K = 30.08, $p < 0.0001$). Significant differences in the spatial distribution of *R. pulmo* were recorded between the TB, NEA, and the rest of the studied areas (CEA, SEA including BK) (K = 191.23, $p < 0.0001$).

A principal component analysis (PCA) was conducted on an abundance data set for four Scyphozoa taxa *Aurelia* spp., *C. tuberculata*, *C. hysoscella* and *R. pulmo*, Ctenophora group and *M. leidyi* during the period from 2010 to 2019 in all investigated areas of the Adriatic Sea (Figure 7). The analysis extracted three factors linking taxa with similar patterns of seasonal distribution and explained 77.68% of the total variability. The first factor relates *to Aurelia* spp. and *C. hysoscella* explains 27.91% variability. The first axis is strongly positively correlated with spring samples, and strongly negatively correlated with samples from summer and especially autumn. The second factor is significantly and positively correlated with the Ctenophora group, *M. leidyi* and *R. pulmo*, and explains 29.63% variability. Until autumn 2015, the observations are mainly negatively correlated with the second axis in all seasons, except for winter 2010, 2011, and 2014. From 2016 onwards, they are mostly positively correlated, with considerably high values from summer 2016 to spring 2017, and summer 2018 and 2019. The third factor explains 20.14% of the total variability and it is positively correlated with *C. tuberculata*. The third axis is highly positively correlated with all summer observations, as well as with autumn observations in 2018 and 2019, while negative correlations were found for winter samples.

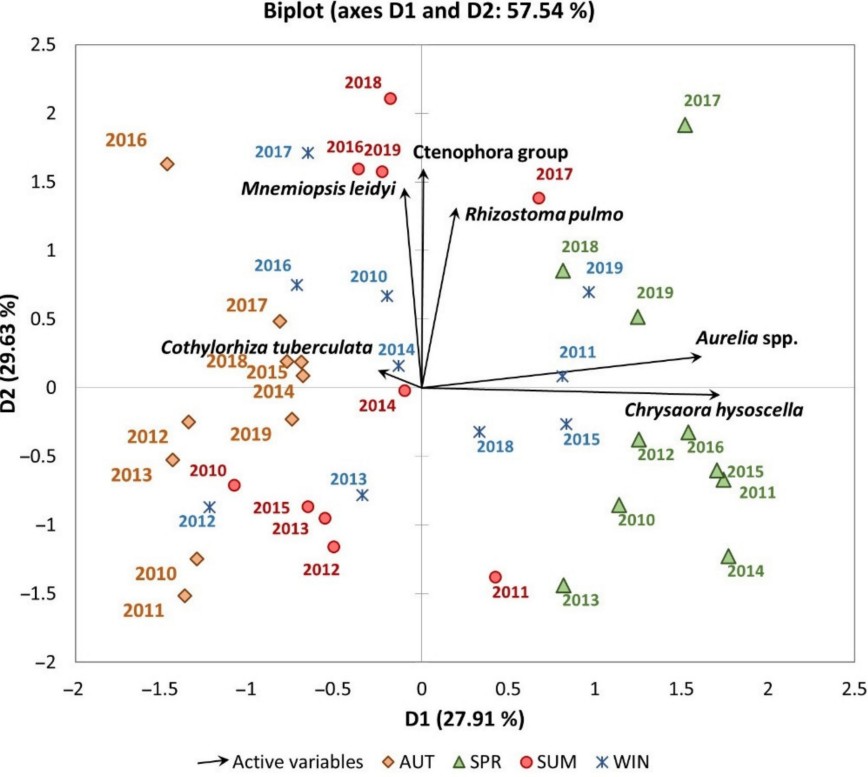

**Figure 7.** Ordering of four Scyphozoa taxa: *Aurelia* spp., *Cotylorhiza tuberculata*, *Chrysaora hysoscella* and *Rhizostoma pulmo*, Ctenophora group: *Bolinopsis vitrea* and *Leucothea multicornis* combined, and *Mnemiopsis leidyi* as active variables obtained by principal component analysis (PCA) during the four seasons from 2010 to 2019 in all investigated areas. Sampling seasons and years are superimposed as active observations and plotted depending on the season as stars (winter), triangles (spring), circles (summer), and rhombus (autumn).

### 3.3. Long-Term Temporal Patterns

There are only very limited time-series data on Scyphomedusae and Ctenophores in the Adriatic. Regular monitoring data of gelatinous macrozooplankton species for several consecutive years are only available from the Gulf of Trieste (our area TB) for 1899–1911. Our classification system for estimating abundance was similar to that used by previous authors [42–51,83] in that period (rare, common, very common). This allowed us to compare our relative abundance and seasonal pattern data with historical ones.

Species regularly observed throughout the 1889–1911 period were: Scyphomedusae *Cotylorhiza tuberculata*, *Rhizostoma pulmo*, *Chrysaora hysoscella*, *Aurelia aurita* and Ctenophores *Eucharis (Leucothea) multicornis*, and genus *Beroe*. The seasonal pattern of most species was remarkably similar to our observations for the following species: *C. tuberculata* was limited to the warm period and peaked in September-October, *C. hysoscella* was rarely abundant but more common in spring, and *Discomedusa lobata* was an uncommon species, recorded in December-February in all but two years. In contrast, we observed *D. lobata* only twice during the period 2010–2019. As in our study, *R. pulmo* was present throughout the year and was classified as common/very common in most years. Similar to our results *L. multicornis* was a common/very common species in the autumn months, less common in other months, and rarest in winter. Contrary to our rare observations, in 1899–1911 *Beroe* spp. has been present all these years, although not abundantly.

The largest discrepancy in jellyfish phenology between our results and those for 1899–1911 is observed in *Aurelia*. In addition to our 2010–2019 data from various areas of the Adriatic Sea and the historical 1899–1911 data from the Gulf of Trieste, we also have observations of *Aurelia* occurrence for the Gulf of Trieste in 1985–1989 and continuously since 1995. In 1899–1911 its occurrence was restricted to January-April, and it rarely occurred in May and December. Moreover, over the entire period, it was rarely classified as common and only twice as very common. A similar pattern was observed in our data until mid-2000, followed by a period of higher abundance and longer duration of the medusa season (Figure 8). In addition, in TB we observed an extensive bloom lasting more than a month during August-September 2020. Similarly, in CEA, the period of *Aurelia* spp. occurrence extended throughout the summer, and mass occurrences were observed in all seasons in 2017/2018 (Figure 4).

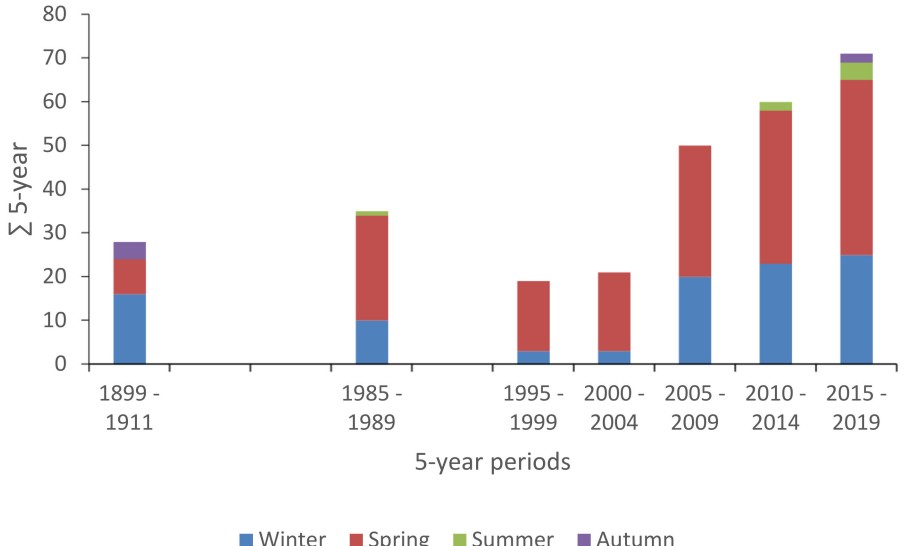

**Figure 8.** Occurrence and abundance of *Aurelia* spp. in different five-year periods in TB.

Records from the early nineteenth century in the Adriatic referred to the species *A. aurita*, following the traditional view of its circumglobal distribution. Mayer [157] also mentioned this species for the Mediterranean Sea, and in the following decades *A. aurita* was considered common in the Mediterranean. However, new molecular techniques have unmasked the cosmopolitan species *A. aurita* as a species complex with several cryptic species [158]. Thus, the identity of the Adriatic *A. aurita* was questioned [75,159]. Finally, Scorrano et al. [77] suggested that, besides the endemic *A. relicta* sp. now occurring only in a saltwater lake on the southern Adriatic island of Mljet, the other two species recently recorded in the Adriatic, *A. coerulea* and *A. solida*, are not native. The former was found only in the Varano lagoon. The introduction of *A. coerulea* into the lagoons seemed to be related to shellfish farming and the importation of shellfish seed. *A. solida* was observed in

several coastal locations in the Mediterranean and the Adriatic seas. At the same time, for *A. solida*, Scorrano et al. [77] suggested migration via the Suez Canal and shipping from the Indian Ocean as the most likely vectors for the introduction. The question of which species was observed by researchers in the nineteenth and early twentieth centuries remains open. Observed changes in *Aurelia* phenology, that is, the extension of the medusa season into the warmest months and the fact that known polyp attachment sites in the Adriatic Sea are restricted to artificial fouled substrates, mainly in harbors and marinas [69,71,160,161], would be consistent with the recent (re)introduction of the warm-water *Aurelia* clade.

## 4. Conclusions

We analyzed phenological data and relative abundances of jellyfish based on scientific surveys and a "citizen science" sighting program along the eastern Adriatic coast during 2010–2019. Inter-annual variability, seasonality and spatial distribution patterns of Scyphomedusae and Ctenophore species were described and compared with existing historical literature. Although the data collected allow only a semi-quantitative assessment of abundance, they provide large-scale spatial and temporal coverage that would otherwise be logistically and financially infeasible. Thus, our empirical analysis of ten years of observations over a large Adriatic area with varying environmental conditions provides information on background variation in Scyphomedusae and Ctenophores populations.

In 2010–2019, the following scyphozoan taxa were found along the eastern Adriatic coast: *Cotylorhiza tuberculata*, *Rhizostoma pulmo*, *Drymonema dalmatinum*, *Chrysaora hysoscella*, *Pelagia noctiluca*, *Mawia benovici*, *Aurelia relicta*, *A. solida* and *Discomedusa lobata*. These Scyphomedusae showed complex spatial and temporal patterns of occurrence and segregation on a spatial and seasonal scale. The temporal pattern of occurrence of meroplanktonic *C. tuberculata*, *C. hysoscella*, *Aurelia* spp. showed distinct seasonality. *R. pulmo* occurred throughout the year with a clear decreasing trend from the northern to the southern Adriatic. The winter-spring dominance characteristic of *Aurelia* has changed recently. *C. tuberculata* typically occurs in summer and autumn, while *C. hysoscella* was weakly represented but most abundant in spring.

Of the eight scyphozoan taxa identified in the 2010–2019 study, mass occurrences with a clear seasonal pattern strongly related to geographic location and the study period were observed for meroplanktonic *Aurelia* spp. and *R. pulmo*, and to a lesser extent for *C. hysoscella*, *C. tuberculata*, *D. lobata*. Holoplanktonic *P. noctiluca* also formed large aggregations that were seasonally less predictable. The highest abundances of Scyphomedusae, with the exception of *P. noctiluca*, were found in the northern Adriatic and decreased towards the southern regions. However, differences between regions have become less pronounced since 2014, mainly due to increased jellyfish abundance during the summer-autumn period (Figure 7; Table 2).

Seven species of Ctenophora were found in our investigations: *Pleurobrachia pileus*, *P. rhodopis*, *Bolinopsis vitrea*, *Leucothea multicornis*, *Mnemiopsis leidyi*, *Cestum veneris*, and *Beroe forskalii*, while large blooms were produced by only three species (*B. vitrea*, *L. multicornis*, *M. leidyi*). The appearance and blooming of *M. leidyi* since 2016 in the northern Adriatic represents one of the most substantial changes to the jellyfish community of the Adriatic Sea in recent years. It contributed to the further differentiation of the northernmost part of the Adriatic from the central and southern parts.

Our results suggest that gelatinous organisms are assuming an increasingly important role in the Adriatic ecosystem, which may alter the balance of the food web and lead to harmful and undesirable effects. This paper presents baseline data on the seasonality of Scyphozoa and Ctenophora species, their blooms and hot-spot areas. However, the major gaps in our knowledge remain data on polyps as crucial components of the scyphozoan life cycle and quantitative data on planktonic phase abundances. Therefore, it is essential to carefully monitor the occurrence of the key jellyfish taxa in the Adriatic Sea, develop models capable of predicting their future trends, and conduct studies focused on determining the

ecological impact of such blooms in a system already disturbed by decades of overfishing, climate change and various forms of pollution.

**Supplementary Materials:** The following are available online at https://www.mdpi.com/article/10.3390/d13050186/s1. Table S1: List of Scyphozoa species in the Mediterranean and the Adriatic seas. Table S2: List of Ctenophora species in the Mediterranean and the Adriatic seas. Figure S3: A standardized annual abundance of the most numerous scyphozoans species along the eastern Adriatic coast during the 2010–2019 period. Note the difference in the y-scale for Aurelia spp. For abbreviations see Study Area section (A = *Cotylorhiza tuberculata*; B = *Rhizostoma pulmo*; C = *Chrysaora hysoscella*; D = *Pelagia noctiluca*; E = *Aurelia* spp.). Figure S4: A standardized annual abundance of *Drymonema dalmatinum* along the eastern Adriatic coast during the 2010–2019 period (blank—jellyfish are not seen at all; 1—sporadic occurrence of individual organisms, yellow dot; 2—frequent occurrence of individual jellyfish specimens and/or small aggregations, blue dot and 3—frequent occurrence of large jellyfish aggregations/blooms, red dot; for abbreviations see Study Area section). Figure S5: Temporal variability (month/year) of relative abundance of *Pelagia noctiluca* along eastern Adriatic coast during 2010–2019 period (blank—jellyfish are not seen at all; 1—sporadic occurrence of individual organisms, yellow dot; 2—frequent occurrence of individual jellyfish specimens and/or small aggregations, blue dot and 3—frequent occurrence of large jellyfish aggregations/blooms, red dot; for abbreviations see Study Area section). Figure S6: Temporal variability (month/year) of relative abundance of *Discomedusa lobata* along the eastern Adriatic coast during the 2010–2019 period (blank—jellyfish are not seen at all; 1—sporadic occurrence of individual organisms, yellow dot; 2—frequent occurrence of individual jellyfish specimens and/or small aggregations, blue dot and 3—frequent occurrence of large jellyfish aggregations/blooms, red dot; for abbreviations see Study Area section). Figure S7: Temporal variability (month/year) of relative abundance of *Bolinopsis vitrea* along the eastern Adriatic coast during the 2010–2019 period (blank—jellyfish are not seen at all; 1—sporadic occurrence of individual organisms, yellow dot; 2—frequent occurrence of individual jellyfish specimens and/or small aggregations, blue dot and 3—frequent occurrence of large jellyfish aggregations/blooms, red dot; For abbreviations see Study Area section). Figure S8: A standardized annual abundance of *Bolinopsis vitrea* (A) and *Leucothea multicornis* (B) along the eastern Adriatic coast during the 2010–2019 period. (for abbreviations see Study Area section). Figure S8: Temporal variability (month/year) of relative abundance of *Leucothea multicornis* along the eastern Adriatic coast during the 2010–2019 period (blank—jellyfish are not seen at all; 1—sporadic occurrence of individual organisms, yellow dot; 2—frequent occurrence of individual jellyfish specimens and/or small aggregations, blue dot and 3—frequent occurrence of large jellyfish aggregations/blooms, red dot; for abbreviations see Study Area section). Figure S9: Temporal variability (month/year) of relative abundance of *Cestum veneris* along the eastern Adriatic coast during the 2010–2019 period (blank—jellyfish are not seen at all; 1—sporadic occurrence of individual organisms, yellow dot; 2—frequent occurrence of individual jellyfish specimens and/or small aggregations, blue dot and 3—frequent occurrence of large jellyfish aggregations/blooms, red dot; for abbreviations see Study Area section).

**Author Contributions:** Conceptualization: D.L. and A.M.; investigation: B.P., M.V., T.K., I.V., and P.P.; literature review: A.M. and D.L., formal analysis: N.B., D.L., and A.M.; writing—original draft preparation: D.L., B.P., A.M., and N.B. All authors have read and agreed to the published version of the manuscript.

**Funding:** This study was financed from Slovenian Research Agency (research core funding P1-0237 and postdoctoral project Z7-1884), Croatian Science Foundation projects "Ecological response of northern Adriatic to climatic changes and anthropogenic impact" (EcoRENA, IP-2016-06-4764), Public enterprise for Coastal Zone Management of Montenegro, Environmental Protection Agency, Montenegro.

**Institutional Review Board Statement:** Not applicable.

**Data Availability Statement:** All data generated or analyzed during this study are included in this article and in the Supplementary Materials.

**Acknowledgments:** We would like to offer sincere thanks to our colleagues Valentina Tirelli, Manja Rogelja, Tihomir Makovec, Marinko Babić, Rade Garić, Ante Žuljević, and Vesna Mačić, whose input and advice significantly improved this paper.

**Conflicts of Interest:** The authors declare no conflict of interest.

**Image Credit:** (Figure 1) (**A**) V. Bernetič, Piran; (**B**) B. Pestorić, Kotor; (**C**) T. Makovec, Piran; (**D**) V. Mačić, Kotor; (**E**) T. Makovec, Piran; (**F**) V. Tirelli, Trst

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
