# Peer review of "Scyphomedusae and Ctenophora of the Eastern Adriatic: Historical Overview and New Data"

_diversity, doi:10.3390/d13050186_

Round 1

Reviewer 1 Report

The manuscript "Scyphomedusae and Ctenophora of the eastern Adriatic: historical overview and new data” by Branka Pestorić, Davor Lučić, Natalia Bojanić, Martin Vodopivec, Tjaša Kogovšek, Ivana Violić, Paolo Paliaga, and Alenka Malej represents an interesting study which aims at compiling species distribution data and answer if those abundances are increasing during the recent past or not. I am aware that it is difficult to decipher jellyfish and ctenophore abundance trends in ecosystems. In that respect the study represents an interesting approach to address this question. However, I have some concern regarding the data presentation and grouping before statistical analyses. The authors take pure sums of confirmed sightings per month and further sum this over the year. However, given this approach, differences in sampling effort directly impact conclusions. In my opinion this biases the conclusions. Therefore, I suggest to change the analyses to an average count per month (instead of pure sum per month) before downstream multidimensional scaling analyses are conducted. Also I would urge the authors to include further variables into their analyses, such as SST temperatures, which are rather easily attained from weather stations. This would considerably strengthen the manuscript and its conclusions.

Additionally, I am surprised that >200 references are cited in this research manuscript, which is not a review, where such excessive citations would be reasonable. Also, nearly 1/4th of the citations refer to the last author. While this can be appropriate e.g. for a review, I miss detailed discussion and key result presentation of the cited work. Example, all studies which investigated the jellyfish community in the Adriatic are named, but it is not specified what those studies actually show and conclude. Even though, large attention is devoted to trends in jellyfish abundance and bloom occurrences, abundance data are not provided (in 99% of the cases), which makes the study not comparable to other regions and therefore of limited impact. In this context I ask the authors to make proper reference to other studies where abundance data have been provided and to include abundance data from their investigation area. Only by including this comparison, the study would be of importance for a broader community. In this context it is also remarkable, that some key references have been missing. I outlined those in the detailed comments.

I am convinced that by addressing those issues this manuscript could be a valuable addition to the literature and of interest for the readership of Diversity.

Detailed comments:

Line 43: “Jellyfish populations can form massive blooms due to life cycles favoured by organisms' traits and local advective transport” The authors might want to add a study showing that those processes are of importance in their investigation area, based on experimental and modelling investigations 1,2. Please modify

Line 54: It is confusing that the authors talk about jellyfish carcasses as the references mostly refer to pelagic tunicate carcasses. Hence a definition of what is lumped into jellyfish should be provided or more specifically taxa differences be considered. This study could be of importance as it considers scyphozoan: 3.

Line 56: Another aspect the authors might want to add (with regard to microbiome characteristics of jellies) is that they harbor a species specific microbiome, including potential fish pathogens, which are of importance for aquaculture activities 4.

Line 67/68: Modify to indirect citation instead of using quotes. Example Mills suggested that only xxxx and provide reference at the end of the sentence again.

Line 105: Here a more appropriate reference including geo-referrenced distribution data is 1.

Line 104/105: Reference order mixed up - reference 40 is referred to later than ref 41. Please delete or reword.

Line 108: reference needs addition “as reviewed in xxx”

Line 121-125: It is nice that all these studies investigated P. noctiluca, but for the reader it would be more informative to give a short compilation about the results. What do these studies investigate, what are the key results… Please modify.

Line 126/127: This statement has been done several times now

Line 131/132: Reference is made to work investigating polyp densities without stating what the key results of those studies represent. Please include information about the cited work instead of just citing work without summarizing the key results and putting those into a context.

Line 142: Change “are the order Coronata” to belong to the order…

Line 152: Following reference is missing (Weiland-Bräuer et al. 2015, Jaspers et al. 2020, Weiland-Bräuer et al. 2020) if you want to add ctenophore microbiome studies, these references will be of importance (Daniels and Breitbart 2012, Dinasquet et al. 2012, Hao et al. 2015, Jaspers et al. 2019, Weiland-Bräuer et al. 2020)

166-171: It would be highly valuable if distribution maps would be included here for visualizations.

Line 183: “relatively shallow” the authors want to add here “for the Mediterranean Sea”, as otherwise it is rather deep for coastal shelf waters.

Line 189-201: The description of the hydrography of the Adriatic Sea needs substantial rewording and additional information. For example, one of the most striking characteristics and primary driving force for structuring the overall species community composition of the Adriatic Sea is its oligotrophic characteristic along the east coast and eutrophic characteristic along the west coast. Eutrophication is due to precipitations and nutrient input via rivers and large cities in Northern areas, which leaves the Adriatic along the Italian coast (due to Coriolis forces). I suggest the authors make a proper account of the hydrography and how this is setting the large scale pattern for species distributions in the area. This can shortly be added as a general introduction before going in depth with the description below.

Methods:

Due to pooling of observations (summing), differences in sampling effort automatically leads to an increase in jellyfish abundances without representing actual abundance increases. Hence the method should be revised and considers number of observations and averages instead of pure pools (sums). Otherwise the results are biased by changes in sampling effort!

Results and Discussion:

It should be possible to include average temperature data (as discussed in line 680 and which should be easily accessible from weather stations) for the respective regions and months. This information would be highly valuable in order to understand changes in population densities between years. Especially since the provided data bear large uncertainty as changes in sampling effort directly relate to increasing abundance recordings due to the data analyses method. I would suggest instead of summing observations, averaging should be conducted to level out some of those uncertainties. More detailed comments are provided below:

As result and discussion section is merged, I would suggest including a small table with all species which have been observed in the region, including reference to first description and maximum abundance observed throughout the records. It seems that the authors have a very good literature overview with more than 200 citations to published papers. However, I miss detailed information about those papers, as it stands right now, it is a pure naming of references without providing key essence from those (as outlined above for several locatons). Also the references are in my opinion biased and primarily include the author´s own contributions. I would suggest focusing on the relevant publications, moving the others to the supporting online information and including a broader geographic range and other relevant studies from the area, such as provided in supporting online table 1 and 2.

Line 325-327: Authors note that Coronate species are not included in the analyses. Then a reference is provided that they occur occasionally. In line 338ff the first species specified is a Coronate one. I would suggest to remove line 339ff and leave coronate species out.

Line 391-394 are confusing. Can this be visualized or approximated in a table with + and – (including color code) for increasing and decreasing abundances in the respective areas as a matrix? This matrix should include all species so one can get an overview from one matrix Fig. this would enhance the readability and comprehend the information which is currently very difficult to grasp.

e.g. Line 438-443 and throughout the ms: It would be valuable to include a more detailed account on how the differently discussed jellyfish populations developed in other areas of the Mediterranean Sea & adjacent waters during the time period in question.

Line 453: locality name is provided without indicating on the map, where this is. Please include in Fig. 1

Line 481: Here comparison to other Mediterranean Sea areas is provided. Please include absolute abundances. Otherwise this information is useless as bloom abundances are subjective and not quantitative.

Line 49-499: Please reword. As it stands now it is misleading. Ballast water is probably the vector?!

Line 520: Communication is not the right wording here. Is connection what you intended to say?

Line 553/554: This only gives meaning if abundance data are provided. Please add.

Line 590/591: Please specify the exact salinity you refer to here.

Line 637. If m-2 abundances are provided, it is important to note the bottom depth as otherwise this information cannot be judged. Hence either re-calculate to m-3 abundances or provide bottom depth information.

Conclusion

It is noted that M. leidyi outnumbered all other jellyfish species during its bloom seasons, but this statement is not substantiated by abundance data and as outlined above, the applied method bears uncertainty with regard to differing sampling effort. Hence this statement can in its current form not be substantiated and needs to be re-formulated to account for those uncertainties or deleted.

Figures:

Need substantial re-structuring. Maybe merge the Figs into 2 separate instead of >10 Figs.  in order to make comparisons between species easier.

Fig. 1: KB and TB are missing on the map but are noted in the legend. Please include those into the map.

Fig. 8: It would enhance the readability if the legends are removed from each Fig. and instead a map with the investigation area in question be added on the left hand side.

References:

>200 references with 43 self-citations to the last author seems a bit out of place for a research manuscript which is not a review paper. Also, excessive references are provided without including key results from the referenced work, hence it becomes odd that so many references are included in this article. Further, it is obvious that other relevant work is not included. Hence I suggest to shorten the reference list to the key publications, include a more balanced view about other work done and provide key results from the referenced work including abundance estimates for the different species in different areas.

Daniels, C., and M. Breitbart. 2012. Bacterial communities associated with the ctenophores Mnemiopsis leidyi and Beroe ovata. FEMS Microbiology Ecology 82:90-101.

Dinasquet, J., L. Granhag, and L. Riemann. 2012. Stimulated bacterioplankton growth and selection for certain bacterial taxa in the vicinity of the ctenophore Mnemiopsis leidyi. Frontiers in Microbiology 3:302.

Hao, W., G. Gerdts, J. Peplies, and A. Wichels. 2015. Bacterial communities associated with four ctenophore genera from the German Bight (North Sea). FEMS Microbiology Ecology 91.

Hosia, A., C. B. Augustin, J. Dinasquet, L. Granhag, M. L. Paulsen, L. Riemann, J.-M. Rintala, O. Setälä, J. Talvitie, and J. Titelman. 2015. Autumnal bottom-up and top-down impacts of Cyanea capillata: a mesocosm study. Journal of Plankton Research.

Jaspers, C., B. Huwer, E. Antajan, A. Hosia, H.-H. Hinrichsen, A. Biastoch, D. Angel, R. Asmus, C. Augustin, S. Bagheri, S. E. Beggs, T. J. S. Balsby, M. Boersma, D. Bonnet, J. T. Christensen, A. Dänhardt, F. Delpy, T. Falkenhaug, G. Finenko, N. E. C. Fleming, V. Fuentes, B. Galil, A. Gittenberger, D. C. Griffin, H. Haslob, J. Javidpour, L. Kamburska, S. Kube, V. T. Langenberg, M. Lehtiniemi, F. Lombard, A. Malzahn, M. Marambio, V. Mihneva, L. F. Møller, U. Niermann, M. I. Okyar, Z. B. Özdemir, S. Pitois, T. B. H. Reusch, J. Robbens, K. Stefanova, D. Thibault, H. W. van der Veer, L. Vansteenbrugge, L. van Walraven, and A. Woźniczka. 2018a. Ocean current connectivity propelling the secondary spread of a marine invasive comb jelly across western Eurasia. Global Ecology and Biogeography 27:814–827

Jaspers, C., L. Marty, and T. Kiørboe. 2018b. Selection for life-history traits to maximize population growth in an invasive marine species. Global Change Biology 24:1164–1174.

Jaspers, C., N. Weiland-Bräuer, M. C. Rühlemann, J. F. Baines, R. A. Schmitz, and T. B. H. Reusch. 2020. Differences in the microbiota of native and non-indigenous gelatinous zooplankton organisms in a low saline environment. Science of the Total Environment 734:139471.

Weiland-Bräuer, N., S. C. Neulinger, N. Pinnow, S. Kunzel, J. F. Baines, and R. A. Schmitz. 2015. Composition of bacterial communities associated with Aurelia aurita changes with compartment, life stage, and population. Applied and Environmental Microbiology 81:6038-6052.

Weiland-Bräuer, N., D. Prasse, A. Brauer, C. Jaspers, T. B. H. Reusch, and R. A. Schmitz. 2020. Cultivable microbiota associated with Aurelia aurita andMnemiopsis leidyi. MicrobiologyOpen 9.

Author Response

Dear reviewer,
many thanks for your effort and advice that have greatly contributed to the quality of our manuscript. We have adopted your remarks as you can see in our answers that follow.

Authors’ response to Report 1

General comments

The manuscript "Scyphomedusae and Ctenophora of the eastern Adriatic: historical overview and new data” by Branka Pestorić, Davor Lučić, Natalia Bojanić, Martin Vodopivec, Tjaša Kogovšek, Ivana Violić, Paolo Paliaga, and Alenka Malej represents an interesting study which aims at compiling species distribution data and answer if those abundances are increasing during the recent past or not. I am aware that it is difficult to decipher jellyfish and ctenophore abundance trends in ecosystems. In that respect the study represents an interesting approach to address this question. However, I have some concern regarding the data presentation and grouping before statistical analyses. The authors take pure sums of confirmed sightings per month and further sum this over the year. However, given this approach, differences in sampling effort directly impact conclusions. In my opinion this biases the conclusions. Therefore, I suggest to change the analyses to an average count per month (instead of pure sum per month) before downstream multidimensional scaling analyses are conducted.

Reply: We are grateful for the reviewer's reference to the difficulties associated with deciphering jellyfish abundances and their trends in ecosystems. We fully agree with the comment that a simple summation of confirmed reports for a given month would be influenced by sampling effort, which would ultimately affect the conclusions drawn from the performed analyses. However, the semi-quantitative monthly values of 0 to 3 presented in this paper, which were used as input data in the multivariate analyses, represent the highest frequency of occurrence of individual jellyfish specimens for a given month, regardless of the number of confirmed sightings. The significance of these numerical values is now described in the Materials and Methods section (lines 278-281). To avoid ambiguity, we have written an additional explanation. (line 281: The value represents the highest frequency of jellyfish occurrence in a given month, regardless of the number of reports received.)

Also I would urge the authors to include further variables into their analyses, such as SST temperatures, which are rather easily attained from weather stations. This would considerably strengthen the manuscript and its conclusions. 

Reply: We appreciate the reviewer's suggestion to include SST, and we agree that such data are not difficult to obtain. However, given the number of Scyphozoa and Ctenophora species, their different environmental preferences, and the significant environmental differences among the sites considered, as well as the interannual variability, we believe this is beyond the scope of this article. In the future, we plan to continue our work with a more detailed treatment of the dominant species and to include temperature and some other environmental parameters in the analysis.

Additionally, I am surprised that >200 references are cited in this research manuscript, which is not a review, where such excessive citations would be reasonable. Also, nearly 1/4th of the citations refer to the last author. While this can be appropriate e.g. for a review, I miss detailed discussion and key result presentation of the cited work.

Reply: As stated in the title and in one of our article objectives, we wanted to provide a historical overview of publications on Scyphozoa and Ctenophora in the Adriatic Sea. In addition to more recent references, this overview includes publications from the 19th and first half of the 20th century, most of which are less well known among researchers. Still, we believe that these references contribute to the knowledge of jellyfishes in the studied area and can help to estimate trends. Overall, this section of our paper (1.2. Historical overview) contained more than half of all cited references. Nevertheless, as requested by the reviewer, we reduced the number of references and added some new relevant publications. The number of references is now 165. We have also made a substantial revision of the text to include the main results of the cited papers.

Example, all studies which investigated the jellyfish community in the Adriatic are named, but it is not specified what those studies actually show and conclude. Even though, large attention is devoted to trends in jellyfish abundance and bloom occurrences, abundance data are not provided (in 99% of the cases), which makes the study not comparable to other regions and therefore of limited impact. In this context I ask the authors to make proper reference to other studies where abundance data have been provided and to include abundance data from their investigation area. Only by including this comparison, the study would be of importance for a broader community. In this context it is also remarkable, that some key references have been missing. I outlined those in the detailed comments.

Reply: The text of Section 1.2 Historical overview of studies on Scyphozoa and Ctenophora in the Adriatic Sea has been modified to include the results and conclusions of the reviewed studies. We did not present abundance and trend data in the review, as there are no such data for the area studied, which is a major knowledge gap. The reviewer provided a list of key references (attached below). Several of these references deal with the microbiota associated with gelatinous organisms. We believe this is an extremely important topic, but it is not the main topic of our article. In our paper, we only briefly mention this topic among the contents of the research conducted in the Adriatic Sea (in Section 1.3 Historical overview). Nevertheless, we have included some of the suggested references in our article.

Dinasquet, J., L. Granhag, and L. Riemann. 2012. Stimulated bacterioplankton growth and selection for certain bacterial taxa in the vicinity of the ctenophore Mnemiopsis leidyi. Frontiers in Microbiology 3:302.

Hao, W., G. Gerdts, J. Peplies, and A. Wichels. 2015. Bacterial communities associated with four ctenophore genera from the German Bight (North Sea). FEMS Microbiology Ecology 91.

Hosia, A., C. B. Augustin, J. Dinasquet, L. Granhag, M. L. Paulsen, L. Riemann, J.-M. Rintala, O. Setälä, J. Talvitie, and J. Titelman. 2015. Autumnal bottom-up and top-down impacts of Cyanea capillata: a mesocosm study. Journal of Plankton Research.

Jaspers, C., B. Huwer, E. Antajan, A. Hosia, H.-H. Hinrichsen, A. Biastoch, D. Angel, R. Asmus, C. Augustin, S. Bagheri, S. E. Beggs, T. J. S. Balsby, M. Boersma, D. Bonnet, J. T. Christensen, A. Dänhardt, F. Delpy, T. Falkenhaug, G. Finenko, N. E. C. Fleming, V. Fuentes, B. Galil, A. Gittenberger, D. C. Griffin, H. Haslob, J. Javidpour, L. Kamburska, S. Kube, V. T. Langenberg, M. Lehtiniemi, F. Lombard, A. Malzahn, M. Marambio, V. Mihneva, L. F. Møller, U. Niermann, M. I. Okyar, Z. B. Özdemir, S. Pitois, T. B. H. Reusch, J. Robbens, K. Stefanova, D. Thibault, H. W. van der Veer, L. Vansteenbrugge, L. van Walraven, and A. Woźniczka. 2018a. Ocean current connectivity propelling the secondary spread of a marine invasive comb jelly across western Eurasia. Global Ecology and Biogeography 27:814–827

Jaspers, C., L. Marty, and T. Kiørboe. 2018b. Selection for life-history traits to maximize population growth in an invasive marine species. Global Change Biology 24:1164–1174.

Jaspers, C., N. Weiland-Bräuer, M. C. Rühlemann, J. F. Baines, R. A. Schmitz, and T. B. H. Reusch. 2020. Differences in the microbiota of native and non-indigenous gelatinous zooplankton organisms in a low saline environment. Science of the Total Environment 734:139471.

Weiland-Bräuer, N., S. C. Neulinger, N. Pinnow, S. Kunzel, J. F. Baines, and R. A. Schmitz. 2015. Composition of bacterial communities associated with Aurelia aurita changes with compartment, life stage, and population. Applied and Environmental Microbiology 81:6038-6052.

Weiland-Bräuer, N., D. Prasse, A. Brauer, C. Jaspers, T. B. H. Reusch, and R. A. Schmitz. 2020. Cultivable microbiota associated with Aurelia aurita andMnemiopsis leidyi. MicrobiologyOpen 9.

Detailed comments

Line 43: “Jellyfish populations can form massive blooms due to life cycles favoured by organisms' traits and local advective transport” The authors might want to add a study showing that those processes are of importance in their investigation area, based on experimental and modelling investigations.

Reply: Done. We cited a paper with results that support these processes based on the experimental and modelling studies.

Line 54: It is confusing that the authors talk about jellyfish carcasses as the references mostly refer to pelagic tunicate carcasses. Hence a definition of what is lumped into jellyfish should be provided or more specifically taxa differences be considered. This study could be of importance as it considers scyphozoan.

Reply: The cited articles by Billet et al. (2006) and West et al. (2009) refer to the scyphozoans Crambionella orsini and Catostylus mosaicus, respectively, while the paper by Lebrato et al. 2012 deals with the broader topic of jelly-falls, which in almost half of the cases includes Scyphozoa in addition to Thaliacea. In our revised text, we added Scyphozoa, and the sentence now reads: Ultimately, the accumulation of jellyfish carcasses, including Scyphozoa, on the seafloor may affect the benthic biota through bacterial oxygen consumption and remineralization processes.

Line 56: Another aspect the authors might want to add (with regard to microbiome characteristics of jellies) is that they harbour a species specific microbiome, including potential fish pathogens, which are of importance for aquaculture activities.

Reply: Done. Jellyfish-associated microbiota, including potential fish pathogens, is indeed an important topic. We have included some references suggested by the reviewer.

Line 67/68: Modify to indirect citation instead of using quotes. Example Mills suggested that only xxxx and provide reference at the end of the sentence again.

Reply: Done.

Line 105: Here a more appropriate reference including geo-referrenced distribution data is 1.

Reply: We have included a reference with geo-referenced distribution data.

Line 104/105: Reference order mixed up - reference 40 is referred to later than ref 41. Please delete or reword.

Reply: Done. We corrected the order of the references.

Line 108: reference needs addition “as reviewed in xxx”

Reply: Done.

Line 121-125: It is nice that all these studies investigated P. noctiluca, but for the reader it would be more informative to give a short compilation about the results. What do these studies investigate, what are the key results…

Reply: As indicated in our reply to the general comments, we have modified the text according to the reviewer's suggestions. 

Line 126/127: This statement has been done several times now.

Reply: Done. The sentence has been deleted. 

Line 131/132: Reference is made to work investigating polyp densities without stating what the key results of those studies represent. Please include information about the cited work instead of just citing work without summarizing the key results and putting those into a context.

Reply: Done. We added the requested information. 

Line 142: Change “are the order Coronata” to belong to the order…

Reply: Done. 

Line 152: Following reference is missing (Weiland-Bräuer et al. 2015, Jaspers et al. 2020, Weiland-Bräuer et al. 2020) if you want to add ctenophore microbiome studies, these references will be of importance (Daniels and Breitbart 2012, Dinasquet et al. 2012, Hao et al. 2015, Jaspers et al. 2019, Weiland-Bräuer et al. 2020)

Reply: Although our article only briefly mentions some research on the jellyfish-related microbiome in the Adriatic Sea, this is not the topic of our article. However, we agree that the references suggested by the reviewer are relevant, and so we listed some listed some.  

166-171: It would be highly valuable if distribution maps would be included here for visualizations.

Copying lines 166 – 171: In any case, most of the jellyfish data along the eastern Adriatic coast relate to the northern Adriatic, and there is much less information for the central and southern parts. Apart from the studies on P. noctiluca blooms and the intensive research of A. relicta in the lake of Mljet Island [90-95], the only records for Scyphozoa and Ctenophora refer to the presence/absence of certain species in these regions [112] and the occurrence of some species in the period 1995-2001 [117]. Reply: We agree that distribution maps would be useful. Unfortunately, there are few, rare data on jellyfish abundances from very different time periods and areas for the Adriatic Sea, so it is currently impossible to produce such maps.  

Line 183: “relatively shallow” the authors want to add here “for the Mediterranean Sea”, as otherwise it is rather deep for coastal shelf waters.

Reply: Done. We have added this clarification in the text. Thank you for this observation. 

Line 189-201: The description of the hydrography of the Adriatic Sea needs substantial rewording and additional information. For example, one of the most striking characteristics and primary driving force for structuring the overall species community composition of the Adriatic Sea is its oligotrophic characteristic along the east coast and eutrophic characteristic along the west coast. Eutrophication is due to precipitations and nutrient input via rivers and large cities in Northern areas, which leaves the Adriatic along the Italian coast (due to Coriolis forces). I suggest the authors make a proper account of the hydrography and how this is setting the large scale pattern for species distributions in the area. This can shortly be added as a general introduction before going in depth with the description below.

Reply: Thank you for the comments on the hydrographic characteristics of the Adriatic Sea and for pointing out the contrast between the oligotrophic character of the southern and eastern Adriatic coast and the eutrophic character of the northern Adriatic, with which we fully agree. The section has been revised and modified accordingly. Additional information has been provided, including the figure showing the general circulation pattern during the studied period.

Methods:

Due to pooling of observations (summing), differences in sampling effort automatically leads to an increase in jellyfish abundances without representing actual abundance increases. Hence the method should be revised and considers number of observations and averages instead of pure pools (sums). Otherwise the results are biased by changes in sampling effort! 

Reply: As noted in our response to the general comments, we fully agree with the reviewer's comment that simply summing the sighting reports for a given month would be affected by sampling effort, which would ultimately affect our conclusions. We also pointed out that the semi-quantitative monthly values from 0 to 3 presented in this paper are not affected by the number of observations, as the monthly data represent the highest frequency of occurrence of individual jellyfish specimens for a given month, regardless of the number of reports received. To avoid ambiguity, we wrote an additional explanation and lines 278-281 in Materials and Methods now read:

Each month of the year is assigned a value between 0 and 3 according to the following criteria: 0 – jellyfish are not seen at all; 1 – sporadic occurrence of individual organisms; 2 – frequent occurrence of individual jellyfish specimens and/or small aggregations; and 3 – frequent occurrence of large aggregations. The value represents the highest frequency of jellyfish occurrence in a given month, regardless of the number of reports received. Several authors have already used similar methods to present the results of macrozooplankton research [4, 77, 156-159].

Results and Discussion:

It should be possible to include average temperature data (as discussed in line 680 and which should be easily accessible from weather stations) for the respective regions and months. This information would be highly valuable in order to understand changes in population densities between years.

Reply: We agree that information on temperature data would be valuable to understand changes in population densities between years. However, it would be difficult to draw meaningful conclusions without more detailed quantitative data on jellyfish abundance and distribution and further in-depth analysis that would require considerable effort. Given the number of different species with variable temperature tolerances/preferences (and in the case of Scyphozoa, we would also need to include polyps), we believe such an analysis is beyond the scope of this article and requires a separate study. In the future, we plan to continue working with a more detailed treatment of dominant species and including more environmental parameters in the analyses.

As result and discussion section is merged, I would suggest including a small table with all species which have been observed in the region, including reference to the first description and maximum abundance observed throughout the records.

Reply: We appreciate this comment and thank you for the suggestion. A table listing all species found in the Adriatic and Mediterranean Sea is already included in the Supplementary Material, while the taxonomic list (including initial description information) of all species observed during our ten-year study period appears at the beginning of the Results and Discussion section. We agree that information on maximum abundance would be very useful. However, apart from the semi-quantitative data in our article, there is very little quantitative data on abundance to date, and even these relate only to individual species in different areas and different time periods. We have therefore highlighted this problem in our conclusions.

It seems that the authors have a very good literature overview with more than 200 citations to published papers. However, I miss detailed information about those papers, as it stands right now, it is a pure naming of references without providing key essence from those (as outlined above for several locatons). Also the references are in my opinion biased and primarily include the author´s own contributions. I would suggest focusing on the relevant publications, moving the others to the supporting online information and including a broader geographic range and other relevant studies from the area, such as provided in supporting online table 1 and 2.

Reply: Taking these and all previous reviewer comments into account, corrections were made by reducing the number of citations, adding relevant papers as suggested by the reviewer, and rewriting and/or adding clarifications to the text of the manuscript whenever possible.

Line 325-327: Authors note that Coronate species are not included in the analyses. Then a reference is provided that they occur occasionally. In line 338ff the first species specified is a Coronate one. I would suggest to remove line 339ff and leave coronate species out.

Reply: Although we agree with this comment, we find any information on this hitherto neglected Scyphozoa group useful.  

Line 391-394 are confusing. Can this be visualized or approximated in a table with + and – (including color code) for increasing and decreasing abundances in the respective areas as a matrix? This matrix should include all species so one can get an overview from one matrix Fig. this would enhance the readability and comprehend the information which is currently very difficult to grasp.

Reply: We believe that the visualisation of the temporal patterns in different areas, as shown in Figure 4, is clear. However, the text has been changed to read as follows:

After low population densities observed from 2010-2014 along the eastern Adriatic coast, detections increased from 2015 to 2017 and then gradually decreased in 2018-2019 (Supplementary Material S3). Mass occurrences were observed very frequently in TB, especially from 2015 to 2018, while outbreaks were less frequent in NEA (Figure 4). C. tuberculata was continuously found in CEA, but outbreaks were only observed in August and September 2017. Bloom was not observed in SEA. In BK, C. tuberculata was generally rarely observed. Mass occurrence was only observed in 2013 and 2019 (Figure 4).

e.g. Line 438-443 and throughout the ms: It would be valuable to include a more detailed account on how the differently discussed jellyfish populations developed in other areas of the Mediterranean Sea & adjacent waters during the time period in question.

Reply: We recognize the importance of comparing data on the abundance of jellyfish populations in the Mediterranean and surrounding seas, so we have added relevant references and distribution data to the revised manuscript when was possible. Taking into account all the above comments, it should be emphasized that data collection and analysis for this group of organisms is extremely challenging, so we believe that this manuscript provides valuable data on the distribution of Scyphomedusae and Ctenophora in the Adriatic Sea and is a suitable basis for further ecological research.

Line 453: locality name is provided without indicating on the map, where this is. Please include in Fig. 1

Reply: The text has been slightly modified, but we have no exact information about the place from which Ernst Haeckel received jellyfish material.

Line 481: Here comparison to other Mediterranean Sea areas is provided. Please include absolute abundances. Otherwise this information is useless as bloom abundances are subjective and not quantitative.

Reply: We fully agree with the reviewer that the use of the term bloom is subjective. However, in most cases, even in publications of the most prestigious journals, this term is used without quantitative data.

Line 49-499: Please reword. As it stands now it is misleading. Ballast water is probably the vector?!

Reply: Done. We have summarized the authors’ description of the cited reference.

Line 520: Communication is not the right wording here. Is connection what you intended to say?

Reply: Corrected.

Line 553/554: This only gives meaning if abundance data are provided. Please add.

Reply: As we have explained, no quantitative data on abundance have been published. However, cited references used the same semi-quantitative methodology as we did, and explicitly described the events as a bloom.

Line 590/591: Please specify the exact salinity you refer to here.

Reply: Done. We have added information about salinity.

Line 637. If m-2 abundances are provided, it is important to note the bottom depth as otherwise this information cannot be judged. Hence either re-calculate to m-3 abundances or provide bottom depth information.

Reply: Done. The depth information has been added.

Conclusion

It is noted that M. leidyi outnumbered all other jellyfish species during its bloom seasons, but this statement is not substantiated by abundance data and as outlined above, the applied method bears uncertainty with regard to differing sampling effort. Hence this statement can in its current form not be substantiated and needs to be re-formulated to account for those uncertainties or deleted. 

Reply: We agree and have deleted the last sentence.

Figures:

Need substantial re-structuring. Maybe merge the Figs into 2 separate instead of >10 Figs.  in order to make comparisons between species easier.

Reply: We agree and thank you for that suggestion. Therefore, we combined separate figures for different species into one. To avoid cluttering the text with two overly large figures for Scyphozoa, we moved them to the Supplementary Material, but added a figure in the same format for Ctenophora.

Fig. 1: KB and TB are missing on the map but are noted in the legend. Please include those into the map.

Reply: Both KB and TB were shown on the map; now we just resized the font to make them more visible.

Fig. 8: It would enhance the readability if the legends are removed from each Fig. and instead a map with the investigation area in question be added on the left hand side.

Reply: Done.

References:

>200 references with 43 self-citations to the last author seems a bit out of place for a research manuscript which is not a review paper. Also, excessive references are provided without including key results from the referenced work, hence it becomes odd that so many references are included in this article. Further, it is obvious that other relevant work is not included. Hence I suggest to shorten the reference list to the key publications, include a more balanced view about other work done and provide key results from the referenced work including abundance estimates for the different species in different areas.

Reply: We believe we have already addressed these concerns, which were expressed at the beginning of the review report in the "General Comments". Regarding the self-citations of the last author, it should be noted that three different persons have the same name. Moreover, the reviewer has counted the name even when it appears in the citation of chapters of books by other persons for which the last author is co-editor. However, we reduced the number of citations where the last author is co-author and included some other citations as requested. In the absence of quantitative data on the abundance of the different species of Scyphozoa and Ctenophora in the Adriatic Sea, we have highlighted this as a large knowledge gap in the conclusions of our article.

Reviewer 2 Report

The  manuscript provides a very valuable database of scyphomedusae and

 ctenophores in the Adriatic Sea.  The text is well written and clear.

 The data is clearly described and the figures effective. Primarily,

 this will be very useful for long term comparisons of trends involving

 these species as well as a guide to local abundance patterns for

 researchers studying these animals.  I find it valuable as it is and I

 favor acceptance to the journal. 

Author Response

Dear reviewer,
Thank you very much for your effort in reviewing our manuscript. Your opinion is particularly important regarding the evaluation of the quality of our manuscript.

Reviewer 3 Report

Dear authors, the paper is good, no changes or clarifications are needed. Just some minor text suggestions into the pdf file.

Author Response

Dear reviewer,

Thank you very much for your effort in reviewing our manuscript. We have included all of the above remarks in the manuscript. Your opinion is particularly important regarding the evaluation of the quality of our manuscript.

Authors

Round 2

Reviewer 1 Report

Thank you for very careful consideration of the review comments and suggestions.